# Circular DNA elements of chromosomal origin are common in healthy human somatic tissue

Henrik Devitt Møller[1], Marghoob Mohiyuddin[2], Iñigo Prada-Luengo [1], M. Reza Sailani[3], Jens Frey Halling[1], Peter Plomgaard[4,5], Lasse Maretty[1], Anders Johannes Hansen[6], Michael P. Snyder[3], Henriette Pilegaard[1], Hugo Y. K. Lam[2] & Birgitte Regenberg [1]

The human genome is generally organized into stable chromosomes, and only tumor cells are known to accumulate kilobase (kb)-sized extrachromosomal circular DNA elements (eccD-NAs). However, it must be expected that kb eccDNAs exist in normal cells as a result of mutations. Here, we purify and sequence eccDNAs from muscle and blood samples from 16 healthy men, detecting ~100,000 unique eccDNA types from 16 million nuclei. Half of these structures carry genes or gene fragments and the majority are smaller than 25 kb. Transcription from eccDNAs suggests that eccDNAs reside in nuclei and recurrence of certain eccDNAs in several individuals implies DNA circularization hotspots. Gene-rich chromosomes contribute to more eccDNAs per megabase and the most transcribed protein-coding gene in muscle, *TTN* (titin), provides the most eccDNAs per gene. Thus, somatic genomes are rich in chromosome-derived eccDNAs that may influence phenotypes through altered gene copy numbers and transcription of full-length or truncated genes.

[1] Department of Biology, University of Copenhagen, Copenhagen DK-2100, Denmark. [2] Roche Sequencing Solutions, Belmont, CA 94002, USA. [3] Department of Genetics, Center for Genomics and Personalized Medicine Stanford University, School of Medicine, Stanford, CA 94305, USA. [4] Department of Clinical Biochemistry, Rigshospitalet, Copenhagen DK-2100, Denmark. [5] The Centre of Inflammation and Metabolism and the Centre for Physical Activity Research, Rigshospitalet, University of Copenhagen, Copenhagen DK-2100, Denmark. [6] The Natural History Museum of Denmark, University of Copenhagen, Copenhagen DK-1350, Denmark. Correspondence and requests for materials should be addressed to H.D.Møl. (email: hdmoller@bio.ku.dk) or to B.R. (email: bregenberg@bio.ku.dk)

The human genome contains 22 linear autosome pairs and a pair of sex-determining chromosomes. This genome structure was previously thought to be highly stable, with minimal divergence between cells[1,2]. Genome-scale sequencing has revealed that insertions, deletions, and amplifications are common in humans[3–5] and even single cells from healthy somatic tissues contain large structural variations[6].

However, the fate of the deleted chromosomal DNA in human somatic cells is unknown. Deleted acentric DNA may be removed from cells by targeted degradation or exclusion from replication. We hypothesized that deleted DNA as well as by-products of damaged DNA can circularize and co-exist in genomes as semi-stable extrachromosomal circular DNA (eccDNA; see Supplementary Note 1 for review of abbreviations for circular DNA). Evidence of eccDNA in humans has existed for more than half a century since double minutes were discovered in tumor specimens from children[7]. EccDNA was later shown to be common in many types of cancers, forming from chromosomal genes and promoting oncogenesis[8–11]. In addition to oncogenes, 5 S ribosomal DNA (rDNA) and repetitive DNA have been found on eccDNA in human cell lines[12–14], showing that a fraction of the genome can exist as eccDNA, at least under certain conditions.

Circular DNA structures of megabase (Mb) sizes are also known as ring chromosomes and can be visualized microscopically by staining metaphase DNA[15,16]. Identification of smaller circular DNA has been by achieved by density separation of DNA by cesium-chloride ultracentrifugation, followed by electron microscopy[17]. EccDNA has also been isolated using linear DNA-specific exonucleases[18,19] or targeted probes for specific eccDNA types and imaging after two-dimensional gel electrophoresis[12] or fluorescence in situ hybridization[10]. Advances in sequencing technologies have allowed for genome-scale identification and mapping of eccDNAs that range in size from 0.1 kilobase (kb) to 2 kb (named microDNA) from human cell lines and mice[20,21]. These <2 kb eccDNAs were shown to derive primarily from genic regions in which exons and 5'-untranslated regions (UTRs) were particularly over-represented[20]. Moreover, recent screens of blood from cancer patients revealed that <2 kb eccDNA can also be found in plasma[22]. The discovery of <2 kb eccDNAs in healthy lung tissue of lung cancer patients suggest that eccDNA either migrates from the malignant tissue or forms in the healthy tissue[22]. In addition, the Wang group recently characterized eccDNAs of up to 20 kb in blood plasma from healthy donors[23]. We hypothesize that the healthy tissue also contains larger eccDNAs of sufficient size to include one or several full-length genes, because (1) eccDNAs larger than 100 kb are found in human tumors[9,10,24], most likely arising by random mutational processes that would be expected to form eccDNAs from all parts of the genome; (2) deletions are known to produce eccDNAs in yeast[25] and in cattle[26]; and (3) deletions in somatic cells can be up to Mb in size[6].

Here, we investigate if eccDNAs are common in somatic tissues from healthy humans (muscle and leukocytes), employing enzymatic removal of linear DNA and sequencing of DNase-resistant eccDNA[27,28]. We detect a large catalog of different eccDNAs formed from chromosomal breakpoints between 0.05 kb and up to 999.8 kb apart. EccDNAs derive from every human chromosome with sequences from all known types of genomic structures, including genes, intergenic, and repetitive regions, revealing that eccDNAs are common mutational elements in human soma. Our discovery suggests that products of deletions are maintained in somatic cells, leading to novel questions regarding their cellular influence and potential biological roles.

## Results

**Genome-wide detection of eccDNAs from soma in healthy men.** To obtain knowledge about eccDNAs in human soma, we adapted the Circle-Seq method[27,28], which detects eccDNAs on a genomic scale, for human tissue (Fig. 1). EccDNAs were purified from skeletal muscle biopsies and blood leukocytes from two groups of healthy men (mean age 62.4 ± 2.4 years). One group ($n$ = 8) had exercised throughout their life (physically active) and the second group ($n$ = 8) had lived a lifelong sedentary lifestyle (physically inactive, Supplementary Table 1). We chose to compare these two groups of men because they have very different lifespan expectancies[29]. Exercise training is reported to protect against oxidative stress and may, therefore, reduce DNA damage, as reported for sperm nuclear DNA from patients with varicocele[30], which ultimately could lead to group differences in eccDNA formation. Our biochemical data confirmed that the oxidative stress levels were significantly higher in muscle tissue from physically inactive men than men with an active lifestyle (protein carbonylation, Supplementary Table 1).

Purification, enrichment, and detection of eccDNAs from somatic samples were performed in four steps (Fig. 1a–c): 1) gentle DNA isolation by column separation, 2) removal of remaining linear DNA by exonuclease, 3) rolling-circle amplification, and 4) sequencing and mapping of paired-end reads to the human genome to identify structural variation resulting from DNA circularization. We confirmed that linear DNA was completely removed after exonuclease treatment using quantitative PCR (qPCR) on a gene absent from eccDNA as a marker (*COX5B*, Supplementary Fig. 1a, b). Each detected circular DNA structure was supported by a minimum of two independent structural-read variants that identified the chromosomal breakpoint coordinates that were joined on the eccDNA (e.g., one split-read and one discordant-read pair, Fig. 1c). In addition, detected eccDNAs were hierarchically ranked based on read coverage by comparison to adjacent upstream and downstream regions. We assigned high confidence (hconf) support to eccDNAs with read coverage >95% and more than two-fold higher mean coverage relative to the summed coverage of adjacent regions of equivalent lengths (Fig. 1c). EccDNAs ranked with confidence (conf) had >95% read coverage, but less than two-fold higher read coverage relative to outside sequences, and finally, eccDNAs with coverage <95% were annotated solely based on the structural-read variants (low quality, lowq) (Fig. 1c, d). In total, we detected 43,960 hconf eccDNAs, 81,066 conf eccDNAs, and 13,655 lowq eccDNAs from muscle samples (average: 2,748 hconf, 5,067 conf, and 853 lowq per sample of $10^6$ nuclei). In leukocytes, we detected 6,253 hconf eccDNAs, 3,191 conf eccDNAs, and 784 lowq eccDNAs (average: 391 hconf, 199 conf, and 49 lowq per sample of $10^4$ nuclei) (Supplementary Table 2). We found saturation tendency in detection of eccDNAs relative to the number of sequenced reads, suggesting that the majority of eccDNAs were recorded by our analysis (Supplementary Fig. 2). Possible false-positive detection was assessed on full-genome human sequence data, the NA12878 Platinum genome data set[31]. When testing the bioinformatics pipeline on NA12878, we found 0 eccDNAs supported by split reads (partial reads mapping to both sides of a junction) and just 54 eccDNAs (0.75 %) supported by soft-clipped reads (partial reads mapping to one side of a junction) (Supplementary Table 2). As 62,763 (45%) and 2,799 (27%) of all detected eccDNAs in muscle tissue and leukocytes were supported by split reads and 53,890 (39%) and 2,325 (23%) eccDNAs were supported by soft-clipped reads, we are confident that the majority of all detections represented true eccDNAs (with an estimated false-positive rate of about 0.75%). In addition, we cannot exclude that some of the signals interpreted as false-positives in the NA12878 genome data

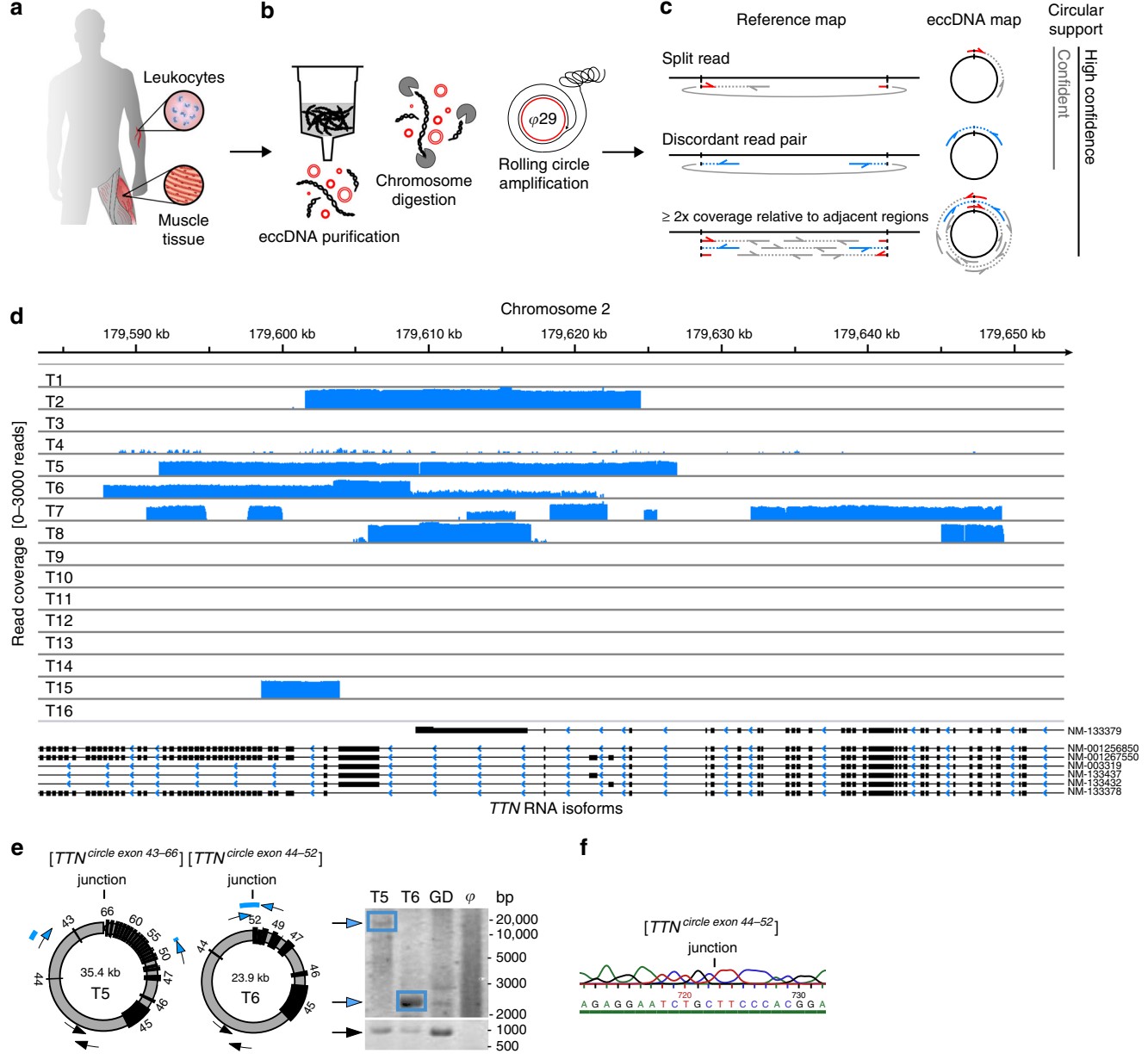

**Fig. 1** Circle-Seq method for mapping of eccDNA. **a** Leukocyte and muscle samples from 16 healthy subjects (T1–T16, $n = 16$). **b** Purification of eccDNA through column separation, exonuclease treatment, and rolling-circle amplification. **c** Detection of eccDNA based on structural-read variants and coverage (soft-clipped, split, red; concordant, gray; and discordant reads, blue). **d** Read coverage display (log-scale) at the titin gene, *TTN*, from muscle samples. **e** EccDNA from T5 and T6 illustrated (black boxes, exons), outward PCR validation (blue arrows), inward PCR (black arrows), and gel-image of T5 and T6 PCR products next to controls: GD, human genomic DNA; $\varphi$, phi29-amplified eccDNA sample without detected [*TTN*$^{circle}$]. **f** Sequence of [*TTN*$^{circle\ exon\ 44-52}$] T6 PCR product at junction

set were actually derived from eccDNA in the whole-genome data. Hence, the rate of false-positives might be lower.

As an example, 130 eccDNAs were mapped to the 0.3-Mb titin gene, *TTN* (Fig. 1d). Two representative examples of *TTN* eccDNAs were 24 kb and 35 kb in size, covering exons 44–52 and 43–66 (Fig. 1e). These two structures were confirmed by outward PCR and Sanger sequencing (Fig. 1e, f) and named [*TTN*$^{circle\ exon\ 43-66}$] and [*TTN*$^{circle\ exon\ 44-52}$] to indicate their genic origin and non-Mendelian, circular character. We apply this nomenclature to all eccDNAs. EccDNAs are also referred to by their chromosomal origin such as [*TTN*$^{circle\ 179,592-179,627kb}$] and [*TTN*$^{circle\ 179,598-179,621kb}$] to accommodate non-genic eccDNA and genic eccDNA without exons (see Supplementary Note 1 for acronyms and nomenclature).

**EccDNAs are common in human soma.** More than 100,000 different eccDNAs ($n = 138,027$) were identified from 16 muscle samples (Supplementary Data 1, Supplementary Fig. 1c-e). All chromosomes were represented in eccDNA sequences, which had a combined length of 12.6% (389.5 Mb) of the human genome. EccDNAs ranked with hconf ranged in size from 0.05 kb up to 57.8 kb with two distinctive peaks at 0.1 kb and 5 kb (Fig. 2a, b). Structures ranked with lower support derived from breakpoints in chromosomes up to 887.9 kb (conf) or 999.8 kb (lowq) apart (Supplementary Data 1), suggesting existence of eccDNAs up to 1 Mb. Although there were individual variations within and between muscle samples from the two groups, the frequency of all eccDNA types showed no significant differences between physically active (median 8,469 different eccDNAs/$10^6$ nuclei, range

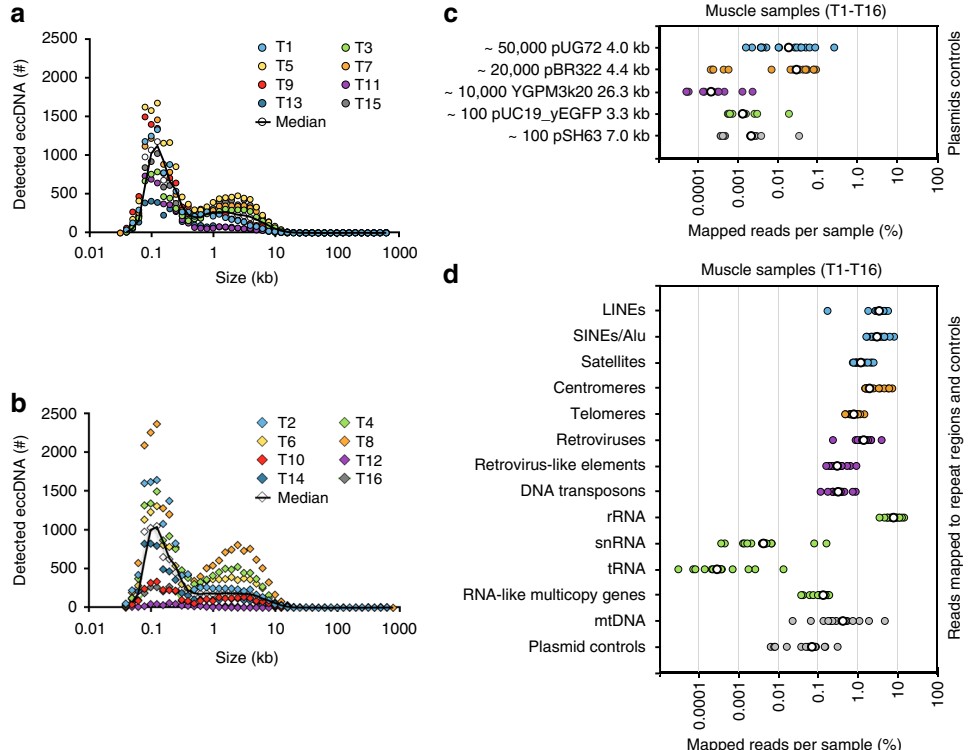

**Fig. 2** EccDNA sizes, read distribution on plasmids and repetitive elements. Number of eccDNAs relative to size in kb for $1 \times 10^6$ muscle nuclei from each individual. **a** Physically inactive men ($n = 8$); **b** active men ($n = 8$). EccDNA kb sizes are in bins of 0.1 log-fold intervals. **c** Percent mapped reads per sample for plasmids added to muscle samples at indicated copy numbers (median, white circles). **d** Repetitive regions from total mapped reads for each muscle-derived eccDNA sample (T1–T16); median, white circles ($n = 16$)

464–19,702) and inactive men (median 9,255 different eccDNA/$10^6$ nuclei, range 3,425–14,498) based on a Wilcoxon rank sum test ($p$-value = 0.95, 95% conf interval, [−7,137; 6,074]). Nevertheless, given the wide conf interval, significant differences between eccDNA counts from inactive and active men might have been missed with the current sample size (Fig. 2a, b). EccDNA from the 16 leukocyte samples, each from ~$10^4$ nuclei (Supplementary Fig. 1f), showed comparable eccDNA frequencies and size distributions (Supplementary Fig. 3a, b, Supplementary Data 2) between physically active (median 356, range 284–919) and inactive men (median 528, range 218–2,347) with no significant differences based on Wilcoxon rank sum tests ($p$-value = 0.43, 95% conf interval, [−783; 136]). When comparing eccDNA frequencies from both types of examined soma, we detected 3.5–5.6 different eccDNAs per 100 nuclei of leukocytes compared to 0.85–0.93 different eccDNAs per 100 nuclei in muscle tissue.

The sensitivity of Circle-Seq was measured using internal plasmid controls added to muscle and blood samples prior to eccDNA purification. Plasmids were recaptured in all samples supporting a detection level of one plasmid per 10,000 nuclei (pSH63, 7 kb, Fig. 2c and Supplementary Fig. 3c). Plasmids larger than 10 kb were also detected, although with fewer mapped reads (e.g., 26-kb YGMP3k20, Fig. 2c). Using plasmid controls, we estimated that Circle-Seq detected ~1–250 eccDNAs per nucleus in the 4-kb range and detected eccDNAs with short interspersed nuclear elements (SINEs) in a broad range of 16–1700 per nucleus. This result is in line with previous reports on HeLa cells with 1600 microDNAs per nucleus in sizes smaller than 400 bp[20] and 50–200 eccDNAs per cell (ref. [32] for review).

**A large fraction of eccDNA reads map to repetitive regions**. Repetitive satellite elements[32] and 5 S rDNA[12] are known to form eccDNAs in human cells. Moreover, repetitive telomeric circles

can lead to human cell immortalization through telomerase-independent elongation of telomeres[33,34]. We found that 0.8% of reads mapped to telomeres, suggesting that [$TEL^{circles}$] are also present in healthy benign tissue. A substantial number of reads also mapped to SINEs (3.1%), long interspersed nuclear elements (LINEs) (3.5%), retroviruses (1.4%), satellites (1.2%), centromeres (2.0%), long terminal repeat (LTR) retrotransposons (0.3%), and rDNA repeats (8.1%) (Fig. 2d; blood Supplementary Fig. 3d), supporting the existence of eccDNAs from all tested types of repetitive elements in the human genome.

**Genic eccDNAs are common**. The majority of eccDNAs (99%) were smaller than 25 kb (Fig. 2a, b, Fig. 3a, Supplementary Fig. 3a, b; Supplementary Fig. 4), while the remaining 1% derived from breakpoints more than 25 kb apart (Fig. 3b). More than half of all eccDNAs came from genic or pseudogenic regions (blood 55%, tissue 52%). For example, the 11.1-kb [$S100A3$-$S100A4^{circle\ 153,512-153,523kb}$] in sample T8 included the complete genes of $S100A3$ and $S100A4$ and parts of $S100A5$. $S100A4$ encodes a protein known to have a causal role in the metastatic spread of tumor cells[35].

**Verification of eccDNAs as large as 35 kb**. We verified 85% of all tested eccDNAs (17 out of 20) by Sanger sequencing of outward PCR products (Fig. 4a, Fig. 1d, f, Supplementary Fig. 4, Supplementary Table 3). Tested unique eccDNAs came both from intergenic and genic loci. For instance, we confirmed the [$ERBB2^{circle\ exon\ 1-7}$], which included the first seven introns and exons of $ERBB2$ (Fig. 4a). This gene encodes the human epidermal growth factor receptor 2 that is associated with cancer when deleted or amplified[8,36,37].

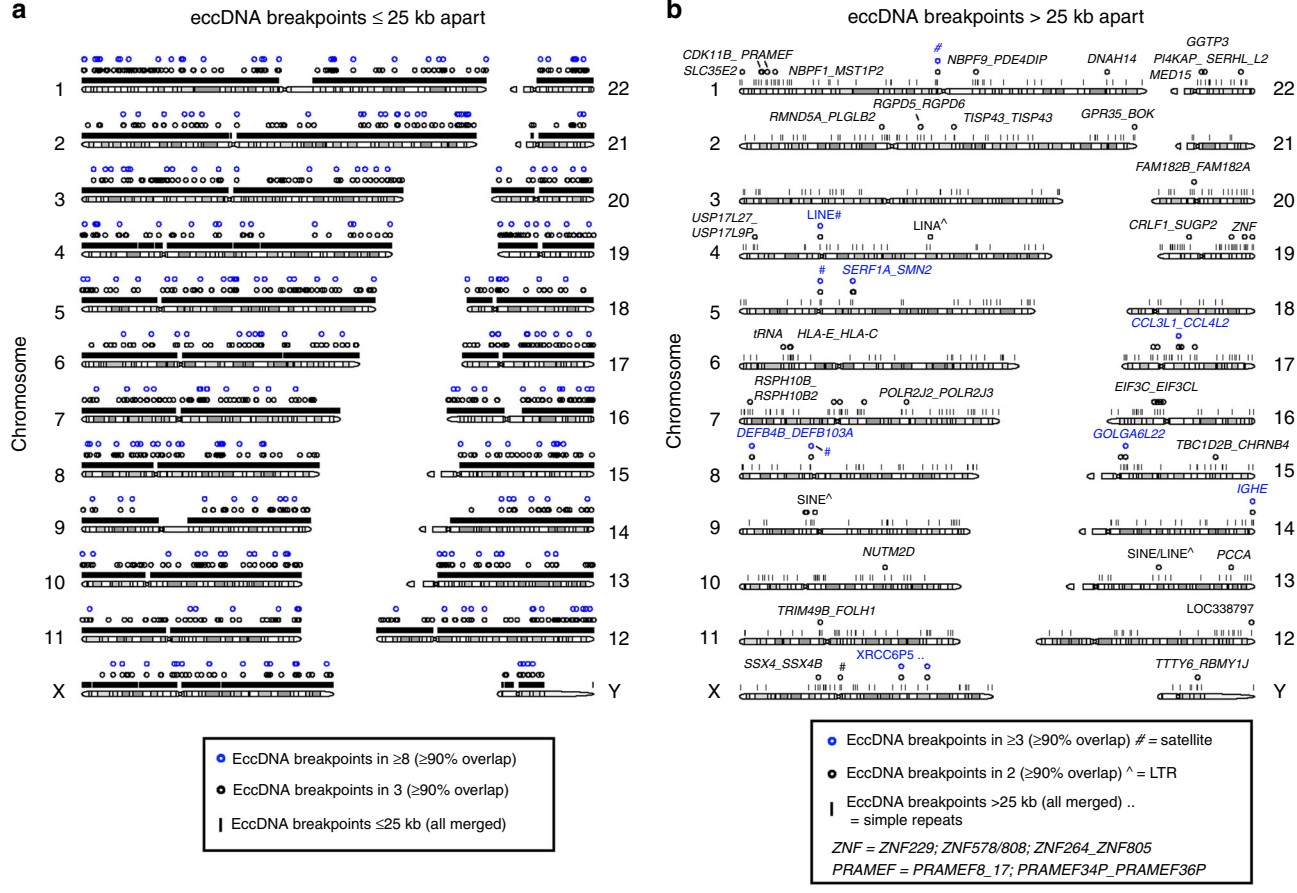

**Fig. 3** Genomic distribution of eccDNA breakpoints. Distribution of detected chromosomal breakpoints across the human genome with read support for circular DNA, in sizes of **a** ≤25 kb (total 147,186) and **b** >25 kb apart (total 1,014) in muscle (T1–T16, $n = 16$) and leukocyte (B1–B16, $n = 16$) samples. EccDNA detected once, black lines; eccDNA detected repeatedly in several individuals with ≥90% overlap, circles

**Deletions correspond to eccDNA.** The discovery of [$ERBB2^{circle\ exon\ 1-7}$] and other eccDNAs with parts of genes suggested that these entities could contribute to the generation of truncated chromosomal and extrachromosomal genes. In one case, we confirmed that a muscle-derived eccDNA also showed evidence of a chromosomal deletion. At the *DAZ4* gene, a DNA deletion was found at the precise site where the [$DAZ4^{circle\ exon\ 18}$] formed (Fig. 4b, c). We also detected eccDNAs from loci reported to frequently undergo gross DNA deletions, such as *HLA* (5 Mb, 6p21.32–p22.1), *KIR* (1 Mb, 19q13.42), and *SERF1A_SMN2* (2.5 Mb, 5q13.2)[38,39] in both blood and muscle (Fig. 3b; Supplementary Data 3). These findings indicated that large deletions can lead to eccDNA in healthy human soma.

**Recurrent breakpoints flank full-length genes.** More than 1000 conf and lowq eccDNAs were derived from chromosomal breakpoints that were more than 25 kb apart (Fig. 3b). With an average gene size of 27 kb in the human genome[40] many of these potential eccDNAs could be large enough to contain full genes. The majority of breakpoints >25 kb apart flanked genic loci (74%), and the largest was [$PRRC2B-SETX^{circle\ 134213-135213kb}$] at 1 Mb (Supplementary Data 3). Notably, several breakpoints >25 kb apart from nearly identical genic or intergenic regions were found in two or more individuals (Fig. 3b). For instance, the 941-kb [$HLA-E\_HLA-C^{circle\ 30,372-31,314kb}$] came from precursor genes of the major histocompatibility complex (class I, E, and C) and was identified in participants B6 and B14. [$SERF1A\_SMN2^{circle\ 68,904-69,760kb}$] was detected by junction reads that mapped to two identical noncoding RNA genes, 838 kb apart, in participants B3,

B5, and B9 (Fig. 3b and Supplementary Data 4). Recurrent eccDNAs are known in yeast, where paralogous genes and repetitive sequences recombine to form identical eccDNAs in independent cell lines[25,27,41]. This phenomenon also appeared to occur in humans as breakpoints in blocks of paralogous genes were found repeatedly in two or more participants. For instance, we found recurrent breakpoints >25 kb apart that flanked genes in the preferentially expressed antigen in melanoma family (*PRAMEF*), the histone cluster 2, defensin betas, the C-C motif chemokine ligands, and killer cell immunoglobulin-like receptors (Fig. 3b and Supplementary Data 3 and 4). Although the potential large eccDNAs had fewer sequence reads assigned to them, the joint probability of finding similar breakpoints by chance in two participants is $<10^{-12}$, using an overlap of 95%. These results suggested that chromosomal breakpoints in loci up to several 100,000 bp apart could lead to eccDNA and some loci were hotspots for DNA circularization.

**Co-occurrence between eccDNAs and structural variants.** We further tested the co-occurrence between coordinates of common structural variants in the human genome[42] and eccDNA breakpoints >25 kb apart. We found 22 hits with a reciprocal overlap of 99%. The hits included, among others, common deletions of genes from *PRAMEF14/15/17/19/20* (220 kb, 1p36.21), *DNAH14* (115 kb, 1q42.12), *PSG4/5/9/10* (191 kb, 19q13.31), as well as repeats from satellites (1p11.2 and 5q11.1) and regions with LINE/SINE/LTR elements (131 kb, 1q12-1q21.1; 35 kb, 15q11.2). We also detected overlap between the 54-kb [$APOL1\_A-POL4^{circle}$] and a common inversion of the same apolipoprotein

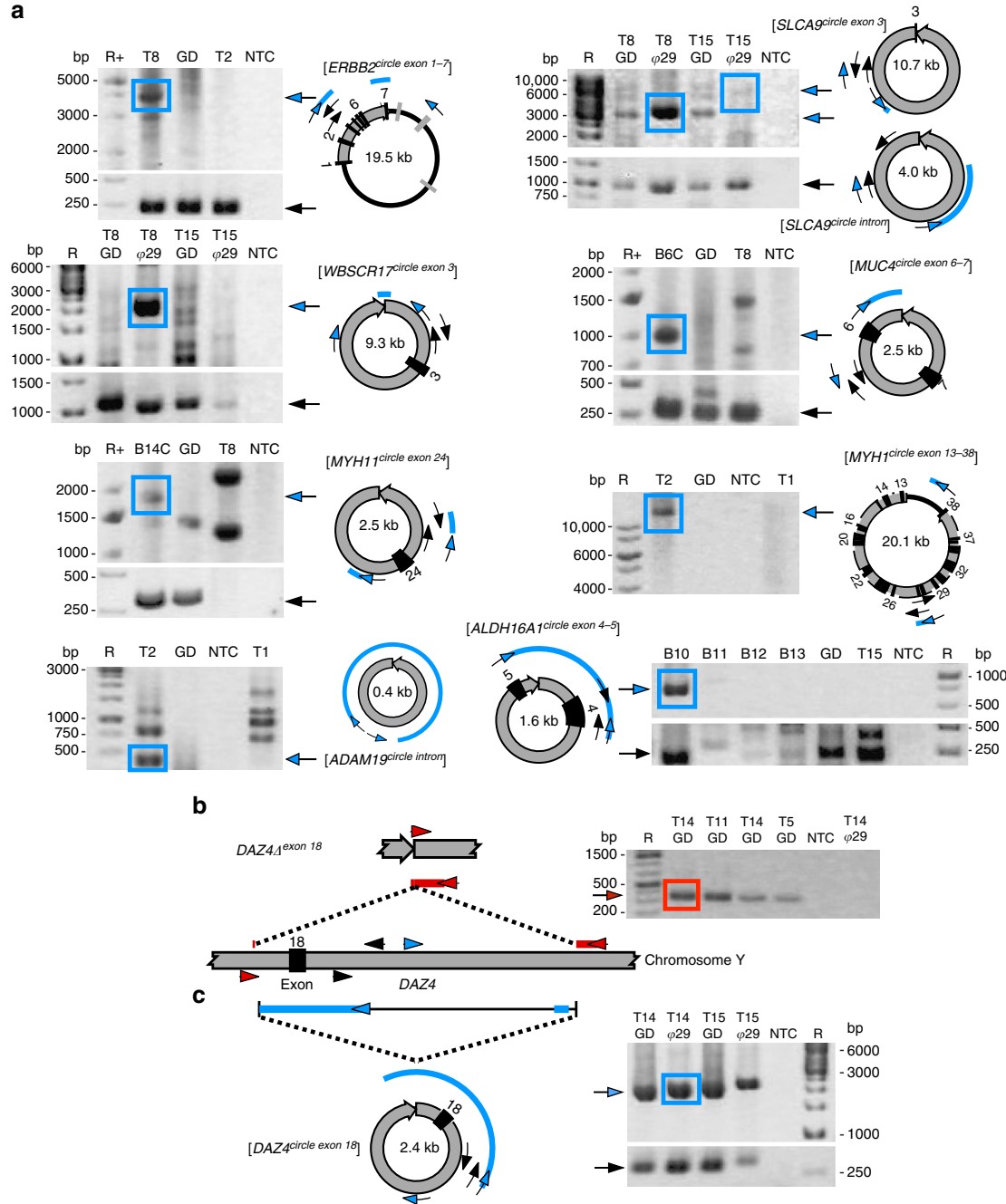

**Fig. 4** EccDNA validation and DNA deletion. **a** Gel images for a validated subset ($n = 9$) of eccDNAs by outward PCR (blue arrows), inward PCR (black arrows), gel electrophoresis, and Sanger sequencing. EccDNAs are named according to gene content; black boxes, exons. Template: T1–T16, muscle B1–B16 leukocyte, phi29 (φ) amplified eccDNA, GD genomic DNA, NTC nontemplate control. Sequenced PCR products are in boxes and alignment of resultant sequence are red and blue lines. **b** Confirmed verification of $DAZ4\Delta^{exon\ 18}$ deletion and **c** [$DAZ4^{circle\ exon\ 18}$] at identical coordinates within the $DAZ4$ locus by inward and outward PCR (oligos arrows, inward red, outward, blue)

genes. Missense variants of the *APOL1* gene are reported to be associated with a 15% increased risk of kidney disease[43]. Finally, a common deletion of the immunoglobulin heavy chain variable region (281 kb, 22q12.3) overlapped an eccDNA detected in sample T6: [$abParts^{circle\ chr2:\ 89,161,023-89,441,956}$].

**Highest frequency of eccDNAs from gene-rich chromosomes**. The genomic distribution of eccDNAs revealed that the gene-rich chromosomes 17 and 19 contributed to a, respective, 1.7-fold and 2.9-fold higher average frequency of eccDNAs per Mb than other

chromosomes (Fig. 5a, Supplementary Fig. 5a). We found positive correlation between the ratios of eccDNA/Mb and coding genes/Mb (Fig. 5b, Supplementary Fig. 5b, $\rho = 0.78$), while pseudogenes, short variants, and long, short, or miscellaneous types of non-coding genes correlated less well with eccDNA frequency ($\rho \leq 0.69$, Supplementary Fig. 5c-g). This result suggested that transcription or other characteristics of coding genes affected the frequency of eccDNA formation. We measured the global mRNA level in all obtained muscle samples to examine whether transcription could explain eccDNA frequencies. Although some of the most highly expressed genes gave rise to many eccDNAs, in

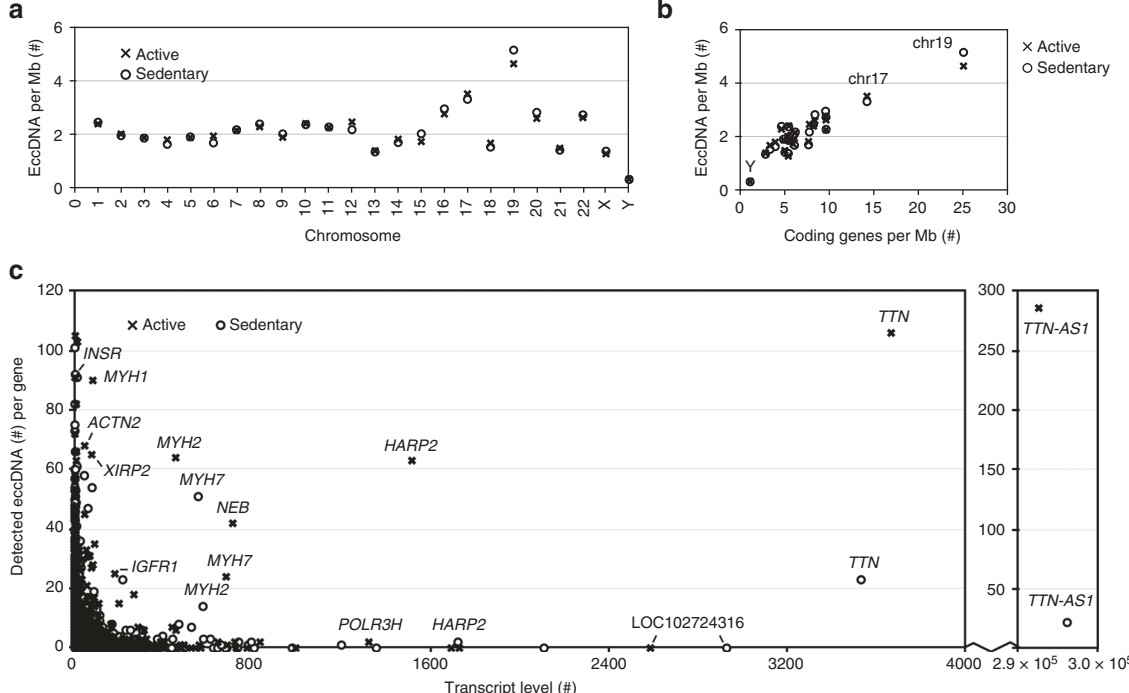

**Fig. 5** EccDNA frequency relative to chromosome, gene density and transcription. Muscle-derived eccDNA counts per Mb from physically active (x, $n = 8$) and inactive men (o, $n = 8$) per: **a** chromosome and **b** coding genes per Mb (chromosome Y, 17 and 19 are marked). **c** EccDNA counts per gene relative to average transcription level of muscle sample (two sets, each group $n = 8$)

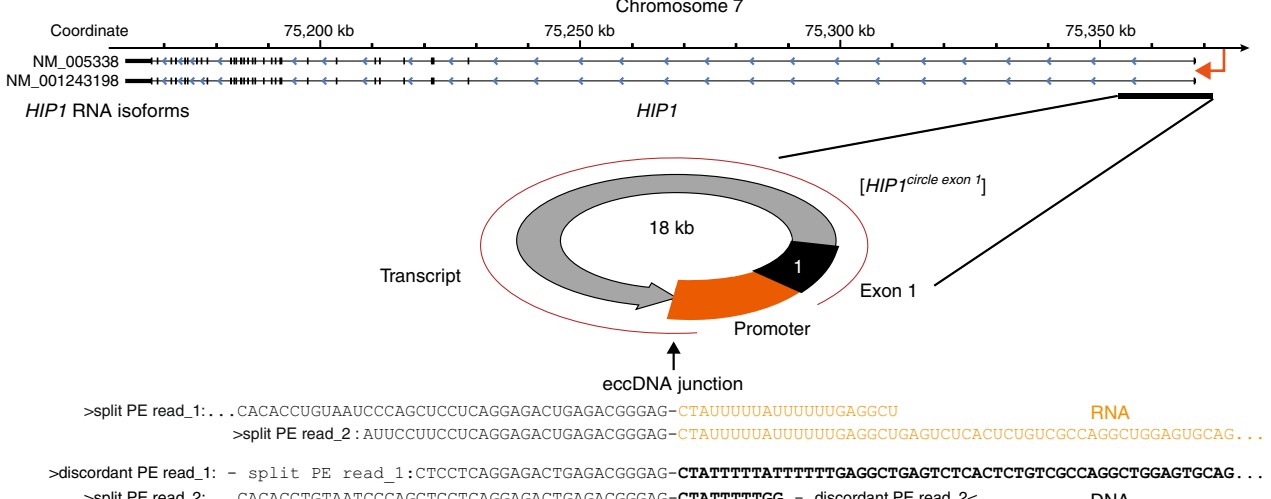

**Fig. 6** EccDNA transcript from the [$HIP1^{circle\ exon\ 1}$]. Display of chromosome 7 at the $HIP1$ gene and detected [$HIP1^{circle\ exon\ 1}$]. The two sequenced RNA reads (black and orange text) from participant T8 overlap perfectly to the two sequenced DNA reads from participant T8 (black and bold text font)

particular the *TTN-AS1* and *TTN* genes of the physically active group, with >100 eccDNAs (Fig. 5c, Supplementary Data 5), we found no general correlation between the numbers of eccDNAs per gene and transcript level.

**Transcription of eccDNA.** To investigate whether eccDNAs were transcribed, we screened mRNA sequences from muscle tissue that could match transcription events across the junction of detected eccDNA. We identified junction transcripts from 25 different eccDNAs in muscle samples from eight participants (Supplementary Table 4, $p < 0.001$, Monte Carlo simulation). The sampled eccDNA transcripts confirmed the existence of eccDNA. For example, in five participants, we detected split-

transcript reads from the *TTN* gene that matched detected [$TTN^{circle}$] coordinates, one of which had a detected size of 612,228 bp. We detected, with hconf, a perfect overlap between eccDNA coordinates of the 18-kb [$HIP1^{circle\ exon\ 1}$] in participant 8 (T8) and mRNA transcription of the first exon of Huntington-interacting protein 1 from T8 (Fig. 6). The transcript data suggested that at least a fraction of the eccDNAs was present in nuclei and not sequestered in transcriptionally inactive compartments of the tissue. The eccDNA transcripts further excluded the possibility that detected eccDNAs were merely by-products of apoptosis-driven fragmentation, as suggested for <2-kb eccDNA (microDNA)[44], or were produced during eccDNA purification.

**EccDNA variation between cells in the same individual**. To determine the diversity of eccDNAs in cells from the same individual, we isolated and sequenced eccDNA from triplicate leukocyte samples from two individuals ($3 \times 10^4$) cells per sample ($n = 3$ for each participant, Supplementary Fig. 6) along with a control for endonuclease digestion of mitochondrial DNA (Supplementary Fig. 6g, h). Samples contained similar numbers and size distributions of eccDNAs (median $2,169 \pm 429$ and median $3,155 \pm 453$; Supplementary Fig. 7, Supplementary Data 6). However, the genetic content of eccDNA was very different between leukocytes from the same individual. From the three samples from participant 6, only 12 eccDNAs had overlapping sequences and only 13 from participant 14 overlapped (Supplementary Fig. 8). We reasoned that the small overlap might result from the constant turnover of leukocytes from the billions of hematopoietic stem cells and myeloid/lymphoid progenitor cells in the human body. Different DNA circularization events in these stem cells would explain a large individual variation between otherwise genetically identical cells.

## Discussion

EccDNA has long been known to form from proto-oncogenes and repeats in the human genome[9,10,32,45,46]. However, the diversity of eccDNA identified in this study supports a model in which any part of the human genome can contribute to eccDNA. Our model implies that deleted DNA is not immediately lost or degraded, but persists in the nucleus as eccDNA, formed through various circularization paths that are likely mediated by the repair machinery for DNA damage, such as nonhomologous end-joining[20,23,24], nonallelic homologous recombination[34], or microhomology-mediated end-joining[20,23]. EccDNAs are common in yeast[27] and plants[47,48], suggesting that the propensity for genomes to form and retain eccDNA is conserved among eukaryotes. Our data further reveal that large parts of the human genome can be found on eccDNA. Although this finding suggests that eccDNAs are formed through random mutational processes, we also found hotspots for eccDNA formation in the human genome. We found that gene-rich chromosomes, repetitive sequences, and tandem paralogous genes had a higher tendency to circularize and form eccDNA, in agreement with previous data from human germline and yeast[25,27,49,50]. Recombination between tandem paralogous sequences lead both to eccDNA and a corresponding deletion in yeast[25]. We also found recurrent eccDNA, and corresponding deletions (Fig. 4c), supporting the hypothesis that certain loci in human somatic tissue have a high propensity to circularize and form chromosomal deletions.

EccDNAs mapped more frequently to gene-rich chromosomes indicating that genic regions generally mutate at higher rates. This result supports previous findings of overrepresented 5′UTRs, exons[20] and 3′UTRs[23] on eccDNA. Of particular interest in this context is that the most highly transcribed gene, *TTN* (encoding titin), also produced the highest number of eccDNAs ($n = 130$, physically active group, Fig. 5). We confirmed the 35.4-kb [*TTN*$^{circle\ exon\ 43-66}$] and 23.9-kb [*TTN*$^{circle\ exon\ 44-52}$] and also found support for [*TTN*$^{circle}$] transcription in five participants (Supplementary Table 4). Titin is expressed in muscle, where it provides elasticity in the resting state. It is the most abundant protein in the human body with an approximate mass of 0.5 kg in an adult human[51]. Loss of titin leads to myopathies[52], making it plausible that the formation of many [*TTN*$^{circles}$] represents a mutational load that reflects high transcriptional activity of *TTN* in the muscle. However, how eccDNA formation frequencies are connected to genic DNA is still unclear. Direct repeat recombination is found to increase upon transcription[53] and transcription itself could provide the basis for DNA damage through R-loops

that consist of a DNA:RNA hybrid, exposing single-stranded DNA. R-loops form naturally during transcription, but can have deleterious effects on DNA integrity[54], which is why they could be a link between transcriptionally active DNA and eccDNA formation. EccDNAs themselves were also transcribed. We found 25 transcripts across eccDNA junctions, suggesting that at least a fraction of eccDNAs resides inside the nucleus. The number of identified eccDNA transcripts is likely an underestimation of the true number, because tissue for eccDNA mapping and mRNA isolation was sampled from different sections of the same biopsies and only eccDNA or mRNA that was present in larger parts of the muscle would be expected to be found in both samples. Gene products from eccDNA transcripts could potentially contribute to the phenotype of somatic cells and tissue as reported in yeast[25,55,56], where for instance, eccDNA with an amino acid transporter gene was selected in cells grown under amino acid limitation[25]. In human cells, eccDNAs with copies of proto-oncogenes, such as *cMYC* and *EGFR*, affect phenotypes by inducing tumorigenesis via increased eccDNA copy numbers, leading to elevated transcript levels[10,57,58].

In the present study, nearly all eccDNAs were acentric and could be expected to segregate nonfaithfully upon replication in mitosis, as previously shown for double minutes in tumors[45]. This phenomenon provides the basis for cell-to-cell variation in both protein expression and protein isoforms. EccDNA might further alter traits if they integrate back into chromosomes as reported in tumor xenografts[11], cattle[26], and wine yeast[59].

We detected more than a thousand different breakpoints more than 25 kb apart. Each of these putative circular DNA structures was detected on the basis of two independent pairs of structural-read variants (Fig. 1c) that both supported the existence of a potential large eccDNA. Investigation of three-dimensional contact domains in the human genome supported the existence of large loop structures from 40 kb to 3 Mb (median size 185 kb), bringing together regions that are far apart[60], and, thus, making circularization plausible. Humans can live with Mb-sized ring chromosomes in their somatic cells[15,16]. In human tumors, double minutes with proto-oncogenes are reported up to 330 kb[9,46], and 28-μm-long eccDNAs (~100 kb) are reported in mouse thymocytes[61]. In this study, some breakpoints >25 kb apart were detected multiple times. However, these breakpoints were not all supported by high internal coverage, presumably because of low abundance and because the phi29 amplification step in the Circle-Seq method is biased for small and more abundant eccDNAs[62]. Moreover, distant breakpoints could resemble more complex structures, as observed for eccDNA in tumor cells[45,63,64]. EccDNAs have also been shown to gradually enlarge through assembly from smaller circular elements[9,45] and perhaps diminish in size over time through internal deletions. Our current eccDNA mapping pipeline does not distinguish between eccDNA derived from a single DNA or several DNA fragments. Therefore, we cannot exclude that some detected eccDNAs resembled complex structures.

In agreement with previous studies[20,22,32], we detected abundant <2 kb eccDNAs (microDNAs), most 100–200 bp. Micro-DNAs are suggested to be products of microdeletions[20] and to form through the mismatch repair pathway[21]. In contrast, a recent study implies that many of these DNA fragments are products of apoptosis[44], requiring further investigation. We cannot rule out that some of the eccDNAs recorded in this study were also products of apoptosis. However, our detection of eccDNA transcripts and multiple recurrent eccDNAs in both blood and tissue samples suggest that eccDNAs are present in living tissue.

In summary, we conclude that eccDNAs are common in healthy human tissue and blood in sizes large enough to carry one

or several complete genes. We find that eccDNAs are transcriptionally active and we suggest that eccDNA contribute to phenotypic variation through expression of full-length and/or truncated genes. Replicating and transcribed acentric eccDNAs are especially expected to have cellular impact, as missegregation in mitosis is expected to yield variable eccDNA copy numbers.

## Methods

**Healthy human subjects.** Two groups of healthy male participants, physically inactive and physically active, were selected based on questionnaires. The inactive group ($n = 8$, age $62.8 \pm 1.3$ years, weight $82.9 \pm 14$ kg) had a lifelong sedentary lifestyle with physical activity once per week at most throughout their life. The physically active group ($n = 8$, age $62.1 \pm 1.4$ years, weight $82.3 \pm 12$ kg) had exercised more than three times per week throughout their life. Ethical approval was granted by the Committee of Copenhagen and Frederiksberg communities, reference number H-7-2014-001 and the research was conducted in accordance with the guidelines of The Declaration of Helsinki. All participants gave their written consent to take part in this study after being informed of the experimental procedures and associated risks. Physiological measurements of percent body fat by dual-energy X-ray absorptiometry scanning confirmed group differences ($30.5 \pm 7.9$ and $21.9 \pm 4.2\%$ in the physically inactive and active groups, respectively). An incremental ergometer cycling challenge consisted of cycling at 120 watts for 5 min, increasing by 20 watts every other minute to perceived exertion of 18 on the Borg scale[65], after which participants continued until exhaustion. The cycling test showed differences in exercise performance between groups, while no significant differences were documented for age, height, weight, cholesterol, testosterone, inflammation, and number of leukocytes (see Supplementary Table 1). Lower protein carbonylation in skeletal muscle from active participants, measured by the OxiSelect ELISA-kit (Cell Biolabs), confirmed a significant lower oxidative stress level in this group compared to inactive participants (Supplementary Table 1).

**Plasmids.** All plasmids were maintained in *Escherichia coli* and purified with a standard plasmid miniprep kit (GeneJet, Thermo Scientific). Plasmid controls were pBR322 (4,361 bp; New England Biolabs), pUC19_yEGFP3 (3,397 bp), pUG72 (3,988 bp; originally pJJH726, EUROSCARF), pSH63 (6,998 bp), and YGPM3k20_pGP564_chrV (26305 bp; Open Biosystems).

**Human samples from healthy human muscle tissue (T1–T16).** Under local anesthesia, muscle biopsies from *vastus lateralis* were collected (Bergström needle), quickly transferred into liquid nitrogen, and later stored at –80 °C. Tissues were fractionated at −20 °C and aliquots of 50–100 mg were sliced into thin pieces with a sterile scalpel and air-dried at room temperature for 1 h before weighing 6 mg tissue (Extended Data Fig. 1a) on an analytical scale (Mettler Toledo). Samples were denoted T1–T16. Odd numbers were used for tissues from the physically inactive group and even numbers for tissues from the physically active group.

**Human leukocytes (B1–B16, B6A–C, B6D; B14A–C, B14D).** Blood (40 mL) was collected from arm veins and centrifuged at 2.6× *g* for 15 min at 4 °C (Eppendorf 5702 R). To match cell concentrations, the middle buffy coat layer (leukocytes) was collected and nuclei were counted (NucleoCounter NC-3000, Chemometec). Samples of ~$10^5$ leukocytes were lysed for 10 min on ice with $1 \times$ buffer C (Qiagen) and $3 \times$ volume ultraclean water and centrifuged at $1,565 \times$ *g* for 15 min at 4 °C. Nuclear pellets were stored at −80 °C until eccDNA purification. The samples were denoted B1–B16. Triplicate aliquots were taken from blood samples of two donors and denoted B6A–C; B14A–C, with two extra samples serving as controls to test potential differences between endonuclease *Not*I (B6D and B14D) and *Mss*I (B6A–C; B14A–C).

**Circle-Seq.** Purification of eukaryotic eccDNA from human somatic tissue was optimized based on the Circle-Seq eccDNA method for budding yeast[28], consisting of multiple steps as described below.

**Cell lysis for Circle-Seq.** In brief, samples of 6 mg tissue ($10^6 \pm 3 \times 10^5$ genomes) or leukocyte pellets ($10^4 \pm 7 \times 10^3$ genomes) were resuspended in 0.63 mL L1 solution (Plasmid Mini AX; A&A Biotechnology) and supplemented with 15 µl Proteinase K (>0.1 U/µl, Life Technologies) before incubation overnight at 50 °C with agitation at 700 rpm (Eppendorf Thermomixer). After cell lysis, a control mixture of plasmids of different sizes and concentrations was added to each sample: 100 copies pSH63, 100 copies pUC19_yEGFP3, 10,000 copies YGPM3k20_pGP564_chrV, 20,000 copies pBR322, and 50,000 copies pUG72. At this point, 30 µl was sampled to assess the input DNA concentration by qPCR.

**Extrachromosomal circular DNA enrichment for Circle-Seq.** Samples were alkaline treated to separate chromosomal DNA, lipids, and protein from eccDNAs by rapid DNA denaturing–renaturing, followed by column chromatography on an ion exchange membrane column (Plasmid Mini AX; A&A Biotechnology). DNA

precipitation was achieved by 45-min incubation at -20 °C (after column elution) and extended centrifugation at 9,788× *g* for 30 min at 2 °C. Precipitated DNA was dissolved in 50 µl water for total DNA 40–419 ng (tissue, T1–T16), 10–224 ng (blood, B1–B16), 22–33 ng (B6A–D) and 36–53 ng (B14A–D) by Qubit dsDNA High Sensitivity assay.

**Removal of linear and mitochondrial DNA for Circle-Seq.** Remaining linear DNA was removed by exonuclease (Plasmid-Safe ATP-dependent DNase, Epicentre), assisted by rare-cutting endonuclease *Mss*I that digested mitochondrial circular DNA (mtDNA, 16 kb) and made additional accessible DNA ends for exonuclease. DNA was treated with two FastDigest Units *Mss*I (GTTT^AAAC) (Thermo Scientific) and incubated at 37 °C for 16 h. The *Not*I (GC^GGCCGC) site is absent in human mtDNA and *Not*I (Thermo Scientific) was used in B6D and B14D blood samples to evaluate use of different endonucleases. Prior to the exonuclease step, endonucleases were thermally inactivated at 65 °C for 10 min (*Mss*I) or at 80 °C for 5 min (*Not*I). Enzymatic reactions with linear-specific exonucleases were at 37 °C in a heating block and chromosomal DNA digestion was carried out continuously for 1 week (144 h tissue, 168 h, blood), adding additional ATP and DNase every 24 h (30 units per day) according to the manufacturer's protocol (Plasmid-Safe ATP-dependent DNase, Epicentre). Finally, the exonuclease was heat inactivated at 70 °C for 30 min. Complete removal of linear DNA was confirmed by qPCR of the *COX5B* gene.

**Rolling-circle amplification of eccDNA for Circle-Seq.** Sample volumes were reduced to 50% under vacuum for 15–20 min (MAXI dry Iyo). Approximately 10% (10 µL) of the total volume of eccDNA-enriched samples was used as template for phi29 polymerase reactions (REPLI-g Midi Kit) amplifying DNA at 30 °C for 2 days (46–48 h).

**EccDNA sequencing from T1–16 and B1–16 for Circle-Seq.** Phi29-amplified DNA was cleaned (QIAquick kits) and sheared by sonication (Bioruptor) to average insert sizes of $430 \pm 30$ nucleotides for tissue and $400 \pm 25$ nucleotides for blood (Bioanalyser). Libraries were prepared by standard methods from ~200 ng purified fragmented DNA, adding adapters and using double indexes (P5 and P7), each with 6-base nucleotide barcodes. The 32 samples were multiplexed in four sets of eight and each set was sequenced as $2 \times 100$-nucleotide paired-end reads on 2 lanes (Illumina HiSeq 2000), collecting up to 200 million paired-end reads/sample (tissue, T1–T16; blood, B1–B16).

**EccDNA sequencing from blood replicates for Circle-Seq.** Each phi29-amplified DNA sample was sheared by sonication to mean fragment size 300 nucleotides (Covaris LE220). The DNA was purified by beads (Ampure bead) and the size distribution of fragments was analyzed (Bioanalyzer QC). The DNA was adjusted to 20 ng/µL, loading 300 ng fragmented DNA for each library preparation (Wafergen's PrepX ILM DNA Library Kit) onto a robotic Apollo 324 system (IntegenX), adding adapters and barcode index labels (6-base oligos). Eight samples (B6A–D, B14A–D) were multiplexed and sequenced as $2 \times 75$-nucleotide paired-end reads on two lanes (Illumina HiSeq 2000 Rapid flowcell) obtaining 30.5–42.5 million paired-end reads per sample for B6A–D and 39.7-50.1 million paired-end reads per sample for B14A–D.

**Deep sequencing of samples T2 and T7 for Circle-Seq.** A pool of two tissue samples (T2 + T7) was resequenced as $2 \times 75$-nucleotide paired-end reads on two lanes (Illumina HiSeq 2500 Rapid) obtaining 650.4 and 255.6 million reads, denoted T2deep and T7deep, respectively.

**Uniquely mapped eccDNA pipeline for Circle-Seq data.** Sequence reads were mapped to a human reference genome to record the origin of chromosomal-derived eccDNAs. The mapping pipeline used aligned structural-read variants to detect eccDNAs and assigned a level of conf to each. Pipeline steps were: 1) Multiplexed datasets were generated by splitting index barcodes, allowing zero mismatches. 2) Adapter sequences and low-quality bases (Q < 10) were trimmed using Cutadapt[66]. 3) To improve eccDNA detection, overlapping sequences in read pairs were merged by SeqPrep (https://github.com/jstjohn/SeqPrep) with default settings to obtain longer singleton contigs. Small eccDNAs with a fragmented DNA insert of <400–500 bp were expected to have substantial overlap between read pairs. 4) The Burrows–Wheeler aligner with maximal exact matches (BWA-MEM)[67] was used to align paired-end reads, singleton contigs, and singleton reads (without a mate) against the GRCh37 human reference assembly. Read alignments were sorted and indexed using samtools[68]. 5) EccDNAs were detected by first annotating chromosomal positions with at least two overlapping structural-read variants (i–iii), i.e., (i) discordant paired-end reads mapped to the opposite orientation (reverse-forward) of the reference genome as opposed to normal concordant mates (forward–reverse), yielding approximate chromosomal coordinates of candidate eccDNAs; (ii) Soft-clipped reads had partially mapped reads (at least 50 soft-clipped bases) at genomic coordinates of potential eccDNA junctions (starts or ends); and (iii) Split reads had both start and end coordinates of putative eccDNA junctions at base-pair resolution (Fig. 1c). Split reads were obtained by remapping

nonmapped fragments of soft-clipped reads using BWA-MEM. If read fragments aligned in the same orientation and to the same chromosome as soft-clipped reads, the chromosomal position was annotated. This mapping used only eccDNAs ≤1 Mb. 6) Next, read coverage of candidate eccDNA-derived regions was assessed by listing mean coverage value with annotations of coverage of adjacent upstream and downstream of regions in the same length as recorded eccDNA. 7) conf levels of eccDNA recordings were assigned as lowq if they had at least two overlapping structural-read variants aligned to a specific genomic location, conf if they also had read coverage of the region of more than 95% and hconf if they satisfied those criteria and mean read coverage was more than twice as high as adjacent regions with equivalent lengths (sum of coverage). 8) EccDNAs that occurred in overlapping regions, but had different sizes were assessed. Although such reads could stem from the same eccDNA molecule, no merging was performed if the reciprocal overlap was less than 50%, as the eccDNA could have derived multiple times from the same locus in close proximity. 9) Final lists of recorded eccDNAs were combined (Supplementary Figure 1, tissue 1-16; Supplementary Fig. 2, blood 1-16; Supplementary Fig. 6, B6A–D + B14A–D), including in all entries details of read support, mean coverage, annotated eccDNA coordinates, and intersections of human genes.

**EccDNA detection as function of mapped reads**. To verify that additional sequencing would lead to saturation in total number of eccDNA detections, we sequenced samples T2 and T7 in depth, obtaining $6.4 \times 10^8$ and $2.5 \times 10^8$ mapped reads in total. Compared to all 16 samples from tissue and leukocytes (Supplementary Fig. 2a–d), deeply sequenced T2 and T7 samples showed a clear tendency to saturation after detection of additional 6,549 and 5,464 eccDNAs, respectively (Supplementary Fig. 2e, f).

**Endonuclease effects on eccDNA counts**. *MssI* has a single site in mtDNA. To evaluate its endonuclease specificity on triplicate blood samples (B6A–C and B14A–C) and potential influence on eccDNA counts and sizes, blood samples from the same donors were also treated with *Not*I (GC^GGCCGC, sample B6D and B14D). MtDNA detection by Circle-Seq and by qPCR was significantly higher in both D samples compared to A–C samples, supporting that *MssI* treatment substantially reduced mtDNA abundance without major impact on eccDNA counts or size distributions (Supplementary Fig. 6g, h).

**Non-unique eccDNA**. Mapping of plasmids and nonunique reads from transposons, centromeres, telomeres, rDNA, satellites, and mtDNA was assessed by mapping all eccDNA-derived reads to a custom genome. The custom genome was based on all consensus sequences from all repeat classes of *Homo sapiens* (583 loci, Repbase, Genetic Information Research Institute) and published sequences from telomeres, centromeres[69], the complete sequence of the human rDNA repeating unit (43 kb, locus HSU13369, accession U13369, PRI 12-MAR-2010), mtDNA (16.57 kb, chrM_GRCh37-hg19), and spiked-in plasmids.

**Quantification of eccDNA based on internal controls**. The number of eccDNAs per nucleus was calculated based on fractions of reads mapped to 4-kb spike-in plasmid controls (pUG72 and pBR322, Fig. 2c). Added to each muscle sample was 50,000 pUG72 and 20,000 pBR322 plasmids. Using the lowest and highest percent read values of pUG72 (0.0015% and 0.2596%) and pBR322 (0.0002% and 0.0905%), we estimated around 1–250 eccDNAs of 4 kb per nucleus. Using percent reads for SINE elements (1.676% and 8.432%) relative to percent reads for pUG72 or pBR322 per nucleus, we estimated [$SINE^{circles}$] of 16–1700 per nucleus with sizes of ~400 bp. These calculations are rough estimates because rolling-circle amplification of eccDNAs with phi29 polymerase is biased toward abundant and small eccDNAs[62].

**Assessment of eccDNA sequencing saturation**. The number of eccDNAs per million reads was determined at different sequence levels by decimation of read amounts at 10% intervals to generate decimation curves (Supplementary Fig. 2).

**EccDNA genomic coverage**. Total coverage of eccDNAs for all samples was found by concatenation and merging of all eccDNAs to calculate the combined length of the genome from which the eccDNA was derived.

**EccDNA per Mb relative to genomic features**. The number of coding genes, noncoding genes, pseudogenes, and short variants per chromosome was obtained from GRCh38.p10 (Ensembl release 88 - Mar 2017 EMBL-EBI) to calculate chromosomal features per Mb relative to the recorded number of eccDNAs per Mb. Correlation tests were calculated using Spearman's rank in R-studio[70].

**Genomic plots**. All intersection operations between eccDNA intervals were set for a reciprocal overlap of at least 90%, using Bedops[71]. Intersection profiles and merged tracks were plotted into a genome map using a modified source code of w4Cseq[72].

**Intersection with common genomic variants**. Breakpoint coordinates of eccDNAs larger than 25 kb were intersected against the database of genomic variants[42] with BedTools (v2.26.0-148-gd1953b6), using a reciprocal overlap of 99%.

**Quantification of genomes per sample**. The number of genomes per sample was quantified by qPCR at the cytochrome C oxidase subunit Vb gene locus (*COX5B*, 2q11.2). To estimate input DNA, 30-μl aliquots, sampled after 18 h of Proteinase K digestion, were used as templates for qPCR after DNA cleaning (magnetic beads, Agencourt). To estimate the remaining linear DNA, 1 week exonuclease-treated DNA was used as template for qPCR. Oligos for qPCR at the human *COX5B* locus were 5' GGGCACCATTTTCCTTGATCAT 3' and 5' AGTCGCCTGCTCTT-CATCAG 3'; at the human mtDNA Cytochrome C oxidase 1 (*MT-CO1*) gene locus 5' GCCCACTTCCACTATGTCCT 3' and 5' GATTTTGGCGTAGGTTTGGTCT 3'. All reactions were run in quadruplicate in a Quant Studio 7 Flex qPCR machine (Applied Biosystems) in 10-μL reactions with 2 μL template (exonuclease-treated samples), 60 nM primers, and 5 μL SYBR Green PCR Master Mix (Applied Biosystems). Reaction conditions were 10 min at 95 °C followed by 40 cycles of 15 sec at 95 °C and 60 sec at 60 °C. To verify reaction specificity, the melting curves were generated and the length of PCR products verified by conventional agarose gel electrophoresis. Purified human genomic DNA in eight serial dilutions was used to produce standard curves for each run. Copy numbers were calculated, assuming two copies of *COX5B* per genome, a molar mass per base pair of 650 g mol$^{-1}$, a genome length of $3.14 \times 10^9$ bp, and $6.77 \times 10^{-3}$ ng DNA per diploid cell. A minimum of two independent qPCR experiments was used for standard deviation calculations. Concentrations of standards were measured using the Qubit dsDNA High Sensitivity assay (Life Technologies).

**Validation of eccDNA recordings**. Outward directing PCR oligos were designed in Primer3web (version 4.0) and devised to yield products across junctions of 20 detected circular DNA structures (Supplementary Table 6). Each 50-μl PCR reaction typically included 120 ng phi29-amplified template (4 μl) or 1 week exonuclease-treated template (4 μL), 200–320 nM primer, dNTP, buffer, and DreamTaq polymerase, and PCR was for 35 cycles in a PCR cycler (Techne PrimeG) under standard PCR conditions. All reactions were performed with controls of human genomic DNA (100–200 ng) and 120 ng from a phi29-amplified sample lacking the eccDNA and nontemplate control. Inward designed oligos (Supplementary Table 6) were positive controls for PCR reactions with circular and linear DNA templates. Size-separation of PCR products on agarose (0.6–1.5%) gel electrophoresis and Sanger sequencing of PCR products confirmed the circular structure of 17 of 20 selected eccDNAs (Fig. 3c, Supplementary Fig. 4). Three eccDNAs missed final Sanger sequencing validation (*HDAC1*, *EGFR*, and *SLC12A8*) and had less coverage and support from structural variation reads than other eccDNAs, indicating they might be rare. In addition to validation of 17 recorded eccDNAs, we validated mitochondrial circular DNA (16 kb) by outward PCR as well as two internal plasmids (pBR322 and pUG72), spiked into prior eccDNA purifications.

**Recording of *DAZ4* deletions**. 5' TGCCTGAAAAGAAAGGTTCCAG 3' and 5' GTCTTAGTGGAACCTTATCACCAG 3' oligos were designed to detect the *DAZ4* deletion in close proximity to the detected coordinates of [$DAZ4^{circle}$], expecting a 310-bp deletion product when using purified genomic DNA as template from participants with [$DAZ4^{circle}$].

**Whole transcriptome sequencing**. For gene expression analyses, total RNA from muscle tissue was extracted using the RNeasy Tissue Mini Kit according to the manufacturer's protocol (QIAGEN). RNA libraries were constructed using the TruSeq Stranded total RNA LT Sample Prep Kit (Illumina) with 500 ng total RNA, according to manufacturer's instructions. RNA-seq libraries were sequenced using the Illumina HiSeq 2000 instrument following manufacturer's instructions. Sequencing was up to 2 × 101 cycles. Image analyses and base callings used the standard Illumina pipeline. The TopHat2 package[73] was used to align reads to the hg19 reference genome, followed by DESEQ2 for RNA expression analysis[74].

**EccDNA transcription detection**. The Segemehl aligner (v0.2.0-418)[75] was used in split-read mode, without realignment, to detect RNA-sequenced split-transcript reads, defined as reads in which the left part aligned to the downstream 3'-end and the right part to the upstream 5'-end. Aligner output was processed with custom Python scripts. Reads aligned to the mitochondrial genome were excluded from downstream analysis. Split-transcript read intervals (similar to "backspliced" reads on circular RNA[76]) that overlapped 100% to detected eccDNA coordinates were collected, discarding split-transcript reads with more than 20-nucleotide deviations from chromosomal start coordinates of detected eccDNAs to eliminate false-positives caused by transsplicing events. Hence, the recording of split-transcript reads was highly restricted to eccDNAs detected with soft-clipped and/or split reads.

Detected eccDNA transcripts in the RNA sequence data were ranked based on the mapping quality of the number of RNA split-transcript reads. Hconf transcribed eccDNA had a perfect match between start and end coordinates with a sequence that was unique in the genome. Conf was similar to hconf, except the

transcript sequence was not unique in the genome. Lowq transcribed eccDNA meant the start or end eccDNA coordinate overlapped the detected transcript and the split-read mapped to multiple positions in the genome.

**Monte Carlo simulations**. The likelihood of detected overlaps between RNA split-transcripts and eccDNA coordinates was assessed by Monte Carlo simulations, using pybedtools[77]. EccDNA coordinates and RNA split-transcript intervals were randomized on the human genome 1,000 times and an empirical p-value was computed based on the number of random intersections relative to the actual detected overlaps between eccDNA and RNA split-transcripts.

**Statistical analyses**. Average and standard deviations were calculated for all data, where Gaussian distribution could be assumed. Median values were used to describe data relating to size, number, and content of eccDNA, as the underlying data distribution was unknown.

**Code availability**. The pipeline for mapping Circle-Seq data was written in Python and the Python package is provided on request for easy installation. It uses multiple CPU cores to ensure fast processing and intermediate files in the pipeline are well annotated to allow detailed analysis of the resulting output. The code is available from Marghoob Mohiyuddin on request.

**Data availability**. Sequence data from Circle-Seq and RNA-Seq experiments have been deposited in the Sequence Read Archive. BioSample accession IDs for DNA Circle-Seq: SAMN08054900 to SAMN08054941. Bioproject ID for RNA-seq PRJNA392413.

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

## Acknowledgements

This research was supported by a research grant (VKR023513) from the Villum Foundation, the Danish Council for Independent Research (FNU 6108-00171B), and the Danish Diabetes Academy. The Centre for Physical Activity Research (CFAS) was supported by TrygFonden. The Centre of Inflammation and Metabolism (CIM) was supported by the Danish National Research Foundation (DNRF55). We would like to thank M. Lisby for large plasmids. Thanks to members of the Regenberg laboratory, J. Ramos Madrigal, T. Gilbert, A. Krogh, L. Parsons, and T. Mourier for fruitful discussions.

## Author contributions

H.D.M. performed experiments with assistance from J.F.H., P.P., B.R., and H.P. M.M built the pipeline for detection of eccDNAs. M.M. and I.P.L. executed bioinformatics analyses with assistance from R.S., L.M., and H.D.M. B.R. and H.D.M. further analyzed the data. H.D.M., H.Y.K.L., H.P., A.J.H., M.P.S., and B.R. designed experiments and the pipeline. H.D.M. and B.R. wrote the manuscript.

## Additional information

**Competing interests:** The authors declare no competing interests.

