## [Peer Review File · Nature Communications]

Reviewer #1 (Remarks to the Author):

This contribution from Moller et al entitled "Circular DNA elements of chromosomal origin are common in the healthy human genome" would have significant merit and general interest once it was filled out and additional experiments presented to support the method and conclusions. At that stage however it would be too long for a Nature Communication article. As it stands there are significant problems and questions which need to be addressed. One would hope that these can be attended to so that the work can appear in a worthy journal.

The single most bothersome issue has to do with the question of how frequent are these circular DNAs in normal nuclei? Perhaps the authors have this data and it was just not clearly enough stated for the reader. What we are told is that some 170,000 circles were detected from 16,000,000 nuclei. This would amount to only one circle in 100 nuclei if the circles are in general single copy elements. Were this the case, then one has to worry that these circles are arising from a 1% population of apoptotic cells present in the normal tissue in which the nuclear DNA is undergoing massive cleavage, degradation and random end to end ligation. Because two amplification methods are being used here (rolling circle replication and PCR) the net result is a highly sensitive method but also one in which unknown variables could generate false positives. It would be important to validate this method by applying it to a number of situations in which cells are known to carry small circular elements of roughly known copy number. This would also provide a base line against which to compare the abundance of these new circular elements. Examples might include ALT cells which contain large numbers of telomeric circles, human cultured cells carrying EBV-based plasmids at roughly 25 copies per cell, or cells carrying AAV vectors.

The authors have not mentioned the studies of Dutta et al (*Science*, 336(6077), 82-6, 2012) in which microDNAs (small double stranded DNA circles) were shown to be present in human cells at up to 100,000 copies per cell. In that study the circles were isolated and directly demonstrated by physical methods including direct visualization. The lower range of circles reported here by Moller and colleagues falls well within the size range of the microDNAs and should have been detected in abundance. If not, then this issue should have been addressed and explanations presented.

As a minor issue, the study of a few number of old Danish men some of whom were more active than others seems to be a weak choice for such a study. Perhaps very young children verses very old adults, healthy individuals verses those undergoing strong chemo-therapy? These would seem to be of more general interest.

Reviewer #2 (Remarks to the Author):

This is a fascinating and potentially important study. The authors adapt their method for examining circular extrachromosomal DNA developed for study in yeast to human muscle tissue and leukocytes. Here they show that small extrachromosomal circular DNA (eccDNA) typically below 25 kb is frequently present in normal cells, and they highlight its potential sites of distribution.

The data will be of considerable interest to readers and the study is novel. The methods are clear and well described and the figures are illustrative. The paper is well written. For the most part, the conclusions are justified by the data.

Two concerns are raised, which if addressed, would greatly improve the MS:

1) The outliers are the large eccDNA containing many genes. These elements have not been independently validated. The conclusion that small circular DNAs are present is not surprising and its confirmation is important. However, the presence of large elements with many genes in normal cells would be very surprising, therefore, additional confirmation would be important to validate

this critical and novel finding.

2) Many muscle cells are a syncytia of many nuclei within a single cell. Does that affect their findings?

3) the method needs to be reproducible by other groups and some parts, the description of the methods needs more description. The authors refer to exonuclease digestion, but unless I missed it, I do not see a mention of which exonuclease was used.

Point-by-point response letter to reviewers

Overall comments to Reviewer #1 and #2

We have revised and extended the entire manuscript addressing all comments made by reviewers and editor. The major changes include:

- Experimental evidence of transcription from eccDNA and thereby also independent evidence of eccDNA (**Figure 6** and Supplementary **Table 9**).
- Modification of the text related to putative eccDNAs large enough to carry entire genes.
- A false positive rate for eccDNA detection.
- Quantification of eccDNA.
- Fulfilling reviews that place our results in full context with previous work, including work from the Dutta laboratory.

Attached are our point-by-point response to critic points raised by Reviewer #1 and Reviewer #2.

Point-by-point response to Reviewer #1

Reviewers' comments (in italic):

Reviewer #1 (Remarks to the Author):

This contribution from Moller et al entitled "Circular DNA elements of chromosomal origin are common in the healthy human genome" would have significant merit and general interest once it was filled out and additional experiments presented to support the method and conclusions. At that stage however it would be too long for a Nature Communication article. As it stands there are significant problems and questions which need to be addressed. One would hope that these can be attended to so that the work can appear in a worthy journal.

Response:

- We thank Reviewer #1 for the revision.

Edits:

- Please see our overall comments and our **comments to Reviewer #2 point 1** on detected >25 kb eccDNA.

1) The single most bothersome issue has to do with the question of how frequent are these circular DNAs in normal nuclei? Perhaps the authors have this data and it was just not clearly enough stated for the reader. What we are told is that some 170,000 circles were detected from 16,000,000 nuclei. This would amount to only one circle in 100 nuclei if the circles are in general single copy elements. Were this the case, then one has to worry that these circles are arising from a 1% population of apoptotic cells present in the normal tissue in which the nuclear DNA is undergoing massive cleavage, degradation and random end to end ligation.

1) Response:

- We agree with Reviewer #1, the previous abstract could be misinterpreted.

- Reviewer #1 may have overlooked data as we added internal plasmid controls to all samples (**Figure 2c, Extended Data Figure 4c**) that can be used for estimating eccDNA/nucleus.
- Detected eccDNAs in the present study were never claimed to be single copy elements and others have shown that eccDNA can replicate.
- We use the internal plasmid controls, added to all samples (**Figure 2c, Extended Data Figure 4c**), to estimate eccDNA/nucleus.

Edits:

- The line “170,000 eccDNAs from 16 million nuclei” in the abstract has been altered: “detecting hundred thousand different eccDNA types from 16 million nuclei”; **page 1 line 24-25** and we state more clearly that the detected eccDNAs are different types of eccDNAs, **page 4, line 5-21**, and **page 8 line 18**.
- We estimate 1 – 250 eccDNAs per nucleus, **page 4, line 28-34**. See more in our edit response to Reviewer #1, point 3.
- We mention that eccDNA can exist in multiple copies and replicate with risk of asymmetric partitioning that can lead to copy number variation as stated **page 8, line 8-14** and **page 9 line 6-8**.

2) Because two amplification methods are being used here (rolling circle replication and PCR) the net result is a highly sensitive method but also one in which unknown variables could generate false positives.

2) Response:

- The Circle-Seq method only uses the high fidelity polymerase phi29 to amplify DNA before eccDNA sequencing.
- We apply to date, the most advanced bioinformatics analysis to justify circular structures, using support from minimum two unrelated paired read variants, i.e. discordant, singleton, split, and soft-clipped reads. We also examine read coverage on putative eccDNA sequences relative to adjacent sequence regions and rank our eccDNA detections (**Figure 1c**).
- We confirm the complete removal of chromosomal DNA by qPCR on *COX5B*, (**Extended Data Figure 2d and page 3, line 16-18**) after 180 U exonuclease, using sequential treatment with ATP-dependent plasmid-safe DNase. That is 36x more units than the Dutta group used (e.g. 1U exonuclease VII per 1 µg/eccDNA + 4U ATP-dependent plasmid-safe DNase per 1 µg/eccDNA, reference¹).
- We validate 17 out of 20 bioinformatic detected eccDNAs by testing the presence of ligated junctions of eccDNA by PCR with outward directing oligo's, before paired-end sequencing and confirm the 17 PCR products by Sanger sequencing (**Figure 1e, Figure 4 and Extended data Figure 4**, the largest tested 35 kb eccDNA, [*TTN^{circle}*], was confirmed).
- We cannot rule out that some of the reported eccDNAs are false positives, so we also ranked eccDNAs according to our level of detection confidence (**Figure 1c, and Supplementary Tables 3-4; 6-7; 9**). The lowest confidence level demands overlap of two independent structural read pairs before annotation of eccDNA with low confidence.
- We now also report a false positive detection rate.
- As mentioned to Reviewer#2, point 1; Hundred large eccDNAs, > 25 kb, were mapped recurrently in several independent participants (**Figure 3b and Supplementary Table 7**). The combined probability that just one of these events arose twice through artifacts or random

chromosomal tandem duplication in independent individuals is 10^{-12} (a 1 kb window). Even with a Bonferroni correction for multiple testing, $p < 10^{-5}$, it is unlikely that the signal is a result of a random event. Hence, large eccDNA is the most parsimonious explanation for these recurrent events.

Edits:

- We now report that the combined probability for finding recurrent > 25 kb eccDNA junctions twice by chance in two participants is less than 10^{-12} , **page 6 line 1-4**.
- A false positive detection rate is now provided (0.75%) after running our pipeline on an unrelated human genomic data (NA12878 Platinum); **page 3, line 27-39** and Supplementary **Table 2**) based on: 0 eccDNAs were supported by split-end reads in the NA12878 Platinum dataset and just only 54 eccDNAs (0.75 %) were supported by soft-clipped reads. We compare the NA12878 Platinum dataset to our eccDNA datasets from tissue and blood on **page 3, line 27-39** and in a new table: Supplementary **Table S2**.

3) It would be important to validate this method by applying it to a number of situations in which cells are known to carry small circular elements of roughly known copy number. This would also provide a base line against which to compare the abundance of these new circular elements. Examples might include ALT cells which contain large numbers of telomeric circles, human cultured cells carrying EBV-based plasmids at roughly 25 copies per cell, or cells carrying AAV vectors.

3) Response:

- This has already been done. We applied and established the Circle-Seq method previously, with strong evidence that eccDNA is common in yeast²⁻⁴. The method is further validated in the current manuscript by the use of 5 spike-in plasmids in different concentrations that were detected in all samples (**Figure 2c** and Extended Data **Figure 4c**).
- Added plasmid controls (**Figure 2c, Extended Data Figure 4c**) are now been used for estimating the number of eccDNA/nucleus.
- Mitochondrial DNA (mtDNA) data from 2x4 leukocyte samples, treated with two different endonucleases MssI and NotI, have also been included. Please note that nuclei pellets were purified from leukocytes (see **page 15, line 26-29**), which means that the majority of mitochondria (with mitochondrial DNA) were lost.

Edits:

- In the new version of the manuscript, we provide an estimate of 1 – 250 eccDNAs per cell in sizes of 4 kb, **page 4, line 28-32** with comparisons to similar numbers in the literature⁵, **page 4, line 32-34** and we added our calculation based on % reads on 4 kb plasmids in methods on **page 19, line 1-11** and approximately 16 to 1700 per nucleus for SINE eccDNAs in sizes of ~ 400 bp (based on % reads on SINEs).
- In the new version of the manuscript, we include text and figures for mtDNA; **page 6, line 40-41**, and Extended **Data Figure 6g-h**.

4) The authors have not mentioned the studies of Dutta et al (Science, 336(6077), 82-6, 2012) in which microDNAs (small double stranded DNA circles) were shown to be present in human cells at up to 100,000 copies per cell. In that study the circles were isolated and directly demonstrated by physical methods including direct visualization. The lower range of circles reported here by Moller and colleagues

falls well within the size range of the microDNAs and should have been detected in abundance. If not, then this issue should have been addressed and explanations presented.

4) Response:

- We agree that the lower range of eccDNA reported in our manuscript falls in the size range of microDNA (see **Figure 2a and 2b**).
- We find comparable concentrations of eccDNA/nucleus. See also answer to Reviewer#1, point 3.

Edits:

- In the new version, we acknowledge previous work⁵, inclusive Dutta and coworkers in the main text⁶. In the new manuscript we reference to microDNA, megabase large circular DNA and provide a historical review of methods applied for eccDNA detection through the last 50 years **page 2, line 5-36**, **page 4, line 32-34** and **page 8, line 38-42**.
- As mentioned to Reviewer#1, point 3; We provide an estimate of 1 – 250 eccDNAs per cell in sizes of 4 kb, **page 4, line 28-32** with comparisons to similar numbers in the literature⁵, **page 4, line 32-34** and we added our calculation based on % reads on 4 kb plasmids in methods on **page 19, line 1-11** and approximately 16 to 1700 per nucleus for SINE eccDNAs in sizes of ~ 400 bp (based on % reads on SINEs).

5) As a minor issue, the study of a few number of old Danish men some of whom were more active than others seems to be a weak choice for such a study. Perhaps very young children verses very old adults, healthy individuals verses those undergoing strong chemo-therapy? These would seem to be of more general interest.

5) Response:

- We have tested 2 x 8 individuals with two tissue types (32 samples) and further tests 2x4 blood samples from two participants. This number of samples is sufficient to show significant differences in many biological systems. The aim of our study is primarily to investigate whether eccDNA exists in a wide range of sizes in healthy tissue.
- We have chosen to compare two groups that are known to have very different life span expectancies (physically active versus a sedentary group), an effect that could have caused differences in oxidative stress and potential differences in DNA damage that ultimately could have led to differences in formation of eccDNA. We have biochemical data that demonstrate significant differences in oxidative stress in the tissue, where the eccDNA was isolated from (protein carbonylation, **Supplementary Table 1**). We have not shown work on cancer patients, as this is the focus by others^{6,7}.
- We have not compared young and old individuals because early studies on eccDNA from mice by Gaubatz and Flores⁸ as their data suggests little difference in the number of two eccDNA repeat types throughout life, and only very young mice carry higher levels of repeat-derived eccDNA.

Edits:

- We have chosen to compare two groups that are known to have very different life span expectancies and have explained our reasoning for having these two groups in the text; **page 2, line 44-47 and page 3, line 1-7**.
- We mention that the biochemical data, **page 3, line 5 and 7** indicate significant differences in oxidative stress in the tissue, from which eccDNA was isolated from (protein carbonylation, **Supplementary Table 1**).

Point-by-point response to Reviewer #2

Reviewer #2 (Remarks to the Author):

This is a fascinating and potentially important study. The authors adapt their method for examining circular extrachromosomal DNA developed for study in yeast to human muscle tissue and leukocytes. Here they show that small extrachromosomal circular DNA (eccDNA) typically below 25 kb is frequently present in normal cells, and they highlight its potential sites of distribution.

The data will be of considerable interest to readers and the study is novel. The methods are clear and well described and the figures are illustrative. The paper is well written. For the most part, the conclusions are justified by the data.

Response:

- We appreciate Reviewer #2 comments.

Two concerns are raised, which if addressed, would greatly improve the MS:

1) The outliers are the large eccDNA containing many genes. These elements have not been independently validated. The conclusion that small circular DNAs are present is not surprising and its confirmation is important. However, the presence of large elements with many genes in normal cells would be very surprising, therefore, additional confirmation would be important to validate this critical and novel finding.

1) Response:

- Evidence for eccDNAs as large as 20.1, 23.9 and 35.4 kb was already provided (**Figure 1e and 4a**).
- In the new version, we provide evidence for transcription from detected eccDNA > 25 kb, supporting the existence of large eccDNA.
- The average human gene is around 25 kb (median 27 kb) with approximately 50% of all human genes that are less than 25 kb in size (source: Guide to the Human Genome (Cold Spring Harbor Lab Press, http://www.cshlp.org/ghg5_all/section/gene.shtml).
- Hundred large eccDNAs, > 25 kb, were mapped recurrently in several independent participants (**Figure 3b and Supplementary Table 7**). The combined probability that just one of these events arose twice through artifacts or random chromosomal tandem duplication in independent individuals is 10^{-12} . Even with a Bonferroni correction for multiple testing, $p < 10^{-12}$.

⁵, it is unlikely that the signal is a result of a random event. Hence, large eccDNA is the most parsimonious explanation for these recurrent events.

- Humans can live with extra copies of megabase large ring chromosomes in their somatic cells¹⁵ and previous reports on double minutes with proto-oncogenes reveal circular elements in the size of 330 kb, e.g. references^{9,10}. Other genes than proto-oncogenes must also be expected on large eccDNAs and because these genes will not induce tumorigenesis, the tissue will have a healthy phenotype.
- Investigation of 3D contact domains in the human genome also supports large loop structures, in size from 40 kb to 3 Mb (median size 185 kb) bringing far-distanced regions in close proximity and thus circularization quite plausible¹¹.
- We confirm the removal of linear DNA by qPCR (**Extended Data Figure 2b**).
- We used our developed bioinformatics pipeline on an unrelated human genomic dataset to estimate the level of possible false positive detections.

Edits:

- In the new manuscript version, we have rewritten and extended the introduction, results, and discussion, emphasizing that we detect chromosomal breakpoints >25 kb apart, which suggests circular DNA structures.
- We report the combined probability for finding recurrent > 25 kb eccDNA junctions twice by chance in two participants is less than 10^{-12} , **page 6 line 1-4**.
- In the new manuscript, we provide additional independent evidence from eccDNA transcription, using our RNA-seq data. This data reveals mRNA transcripts matching eccDNA coordinates, of which 5 eccDNA transcripts were found from the *TTN* gene. The largest transcript was 34.4 kb from a potential 618,228 kb [*TTN^{circle}*]. A 18 kb transcript was detected with high confidence, having a perfect match to the 18 kb [*HIP1^{circle}*], **Figure 6 and page 6 line 28-33**.
- On **page 5, line 5-8 and line 30-31**, we have rewritten the text to clarify that some detected eccDNAs are large enough to carry full genes, e.g. the 11.1 kb [*S100A3-S100A4^{circle 153,512-153,523kb}*].
- Information about average size gene was added “an average gene size of 27 kb in the human genome¹²”, **page 5, line 30-31**.
- In the discussion, **page 8, line 18-37**, we point out that we cannot exclude that some large eccDNAs could potentially contain additional mutations that have changed their size.
- Large eccDNAs and ring chromosomes are mentioned and referenced in the discussion; **page 8, line 24-27**.
- As mentioned to Reviewer#1, point 2; A false positive detection rate is now provided (0.75%) after running our pipeline on an unrelated human genomic data (NA12878 Platinum); **page 3, line 27-39 and Supplementary Table 2**).

2) *Many muscle cells are a syncytia of many nuclei within a single cell. Does that affect their findings?*

2) Response:

- This was already addressed, as blood and tissue show fairly similar eccDNA frequency rates and sizes.

Edits:

- The result is now more explicitly addressed in “*EccDNAs are common in human soma*”, **page 4, line 19 and 22**. “Similar eccDNA frequencies were obtained from both types of examined soma, detecting 3.5 - 5.6 different eccDNAs/100 nuclei of leukocytes compared to 0.85 - 0.93 different eccDNAs/100 nuclei in muscle tissue”.

3.1) *the method needs to be reproducible by other groups and some parts, the description of the methods needs more description.*

3.1) Response:

- We fully agree.

Edits:

- In the new manuscript, we have added a more detailed description. For changes to methods on **page 14-21**.

3.2) *The authors refer to exonuclease digestion, but unless I missed it, I do not see a mention of which exonuclease was used.*

3.2) Response:

- We have used an ATP-dependent plasmid-safe DNase from Epicentre, most likely exonuclease I but the company do not provide description of the exonuclease.

Edits:

- The used DNase is described in methods on “*Removal of linear and mitochondrial DNA*”: **page 16, line 12**.

References

1. Kumar, P. *et al.* Normal and Cancerous Tissues Release Extrachromosomal Circular DNA (eccDNA) into the Circulation. *Mol Cancer Res* (2017). doi:10.1158/1541-7786.MCR-17-0095
2. Møller, H. D., Parsons, L., Jørgensen, T. S., Botstein, D. & Regenbreg, B. Extrachromosomal circular DNA is common in yeast. *Proceedings of the National Academy of Sciences* **112**, E3114–E3122 (2015).
3. Møller, H. D. *et al.* Formation of Extrachromosomal Circular DNA from Long Terminal Repeats of Retrotransposons in *Saccharomyces cerevisiae*. *G3* **6**, 453–462 (2016).
4. Møller, H. D. *et al.* Genome-wide Purification of Extrachromosomal Circular DNA from Eukaryotic Cells. *JoVE* 1–8 (2016). doi:10.3791/54239
5. Gaubatz, J. W. Extrachromosomal circular DNAs and genomic sequence plasticity in eukaryotic cells. *Mutat. Res.* **237**, 271–292 (1990).
6. Shibata, Y. *et al.* Extrachromosomal microDNAs and chromosomal microdeletions in normal tissues. *Science* **336**, 82–86 (2012).
7. Turner, K. M. *et al.* Extrachromosomal oncogene amplification drives tumour evolution and genetic heterogeneity. *Nature* **543**, 122–125 (2017).
8. Gaubatz, J. W. & Flores, S. C. Tissue-specific and age-related variations in repetitive sequences of mouse extrachromosomal circular DNAs. *Mutat. Res.* **237**, 29–36 (1990).
9. Storlazzi, C. T. *et al.* Gene amplification as double minutes or homogeneously staining regions in solid tumors: Origin and structure. *Genome Res.* **20**, 1198–1206 (2010).
10. Albertson, D. G. Gene amplification in cancer. *Trends in Genetics* **22**, 447–455 (2006).

11. Rao, S. S. P. *et al.* A 3D map of the human genome at kilobase resolution reveals principles of chromatin looping. *Cell* **159**, 1665–1680 (2014).
12. Venter, J. C. *et al.* The sequence of the human genome. *Science* **291**, 1304–1351 (2001).

Reviewer #1 (Remarks to the Author):

This is a much improved paper in which the authors have addressed all of the concerns which this reviewer had raised. The work is of general interest and will contribute to the long history of circular DNAs in higher eukaryotic cells. Exactly what mechanisms are involved in the generation of these circles however remains for future studies.

Two minor points: On line 323 the sentence ends without an ending

On line 365 the authors state that circular DNA below 150 bp must be single stranded. This is quite wrong as there are ample examples of double stranded circles in the 70 bp size range (see papers of Charles Richardson in which mini circles are used as DNA templates. The authors just need to delete this comment.

Reviewer #2 (Remarks to the Author):

This reviewer continues to believe that this is a potentially important paper. However, the reviewers have still not fully addressed one of the key concerns, which is proving the existence of some of the larger gene containing eccDNAs using a non-PCR, non sequencing based methods, but rather providing some type of physical evidence. If the authors were to provide such evidence, or alternatively soften the claim if they can't do so, the paper would be stronger.

Reviewer #3 (Remarks to the Author):

This paper describes the identification of extrachromosomal circular DNA molecules in human muscle cells and leukocytes. The paper uses a previously established strategy for identification of eccDNA molecules, named Circle-seq, which essentially purifies and amplifies circular DNA. By paired-end sequencing the circular DNA molecule junction points are mapped.

A main concern appears the novelty of the findings. The authors and others have already published several papers on this topic, thus the identification of eccDNAs *per se* is not new.

The authors do extend previous work by describing somewhat larger eccDNAs up to 25 kb, showing transcription from eccDNAs and providing evidence for hotspots.

Criticisms:

-title: the title suggests that the eccDNAs are part of the human genome, which is not the case, they are derived from chromosomal sequences but not part of it. I would suggest that the authors considering rephrasing this to e.g. Circular DNA elements of chromosomal origin are common in normal human cells. Or something along these lines

-results first section: please consider building this up in a better way. Now it is unclear how many unique eccDNA species were found, how many are recurring in independent samples/individuals, how much filtering was done to get to a robust collection of eccDNA sequences, how many were high and low conf, etc. Please try to be as transparent as possible.

-The classification of eccDNAs in hconf and conf and lowq categories appears somewhat subjective. At the very least the authors should perform experimental validation using e.g. PCR and Sanger sequencing to verify the presence of the eccDNA molecules. How this should be done is unclear though, as many sequences appear unique and can only be verified in amplified DNA. Perhaps the authors could consider a long-read sequencing platform to capture entire circles in reads (i.e. Nanopore sequencing).

-An assessment of data quality was done by testing the NA12878 platinum genome dataset. The authors did not find rigorous support for eccDNAs in these data. Yet, I would suggest that such a dataset should also contain eccDNAs, that were extracted along with the chromosomal DNA when these data were generated. How do the authors explain the complete absence of eccDNAs in the NA12878 data? Further, the authors later on state the presence of 7158 putative eccDNAs in NA12878. This is confusing and should be rephrased or presented in a more insightful way. E.g. consider showing the same statistics as for the muscle and leukocyte samples, i.e. visualize hconf, lowconf eccDNAs, perhaps normalized for the total amount of reads generated. Another point relates to the value of using NA12878 as a measure of specificity. Ideally, the authors should use Phi29 amplified genomic DNA for this. It is well known that different types of rearrangements (inversions/tandem duplications) are induced by Phi29 amplification, which could result in spurious split mappings.

-Another – in my view – important analysis lacking from the paper is a comparison of eccDNA junction points with known SVs breakpoint junctions in the human genome, such as those found in 1KG data. Is there any overlap between such datasets?

-Figure 1e: GD appears unexplained. Further, the T5 band was apparently not sequenced, while the main text states that both T5 and junctions were sequenced. Please clarify.

-Line 152: how many lowq large eccDNAs were experimentally confirmed?

-Line 156: it appears almost impossible from figure 2A/B to find out if there is a difference between the active and inactive groups. How was lack of significance tested? Perhaps a power calculation would be relevant in this context, to at least indicate the smallest difference which could still be picked up by comparing the data from these two groups.

-Line 161: EccDNA from...active and inactive men. This sentence mixes up two different things: the comparison in size distr between leukocytes and muscle, and (ii) differences between active and inactive men. Are these somehow related or are the authors just trying to squeeze two unrelated pieces of information in one sentence?

-Plasmids were used as spike-in controls to measure the sensitivity of Circle-seq: The text describing these results appears rather imprecise. Was the sensitivity to be able to pick up circular DNA molecules 1/10,000 irrespective of circle size? Why is this semi-quantitative? The reported ranges appear very broad, how precise and reliable are these estimates?

-Line 180: the authors state that 0.8% of sequences were mapping to telomeres and conclude that TELcircles are therefore common in normal human cells. But how common is common in this case? 0.8% seems rather uncommon to me. And what is the evidence that these circles containing telomeric sequences are comparable to those described previously (refs 38-40)?

-Line 181: were the fractions of reads from circles that map to LINES/SINEs/etc higher or lower than would be expected on the frequency of these elements in the human genome? I.e. was there over- or underrepresentation?

-Fig4: The authors here attempted to verify several (large) circles with genes. It appears that not all functions were fully recovered in the Sanger sequencing the PCR products. Could the authors clarify the exact confirmation rate and, if the breakpoint junction was verified at nucleotide precision? The title of this section appears somewhat misplaced (lines 188-202) as the verification also involves many smaller circles.

-Line 262: 612,228 kb should probably be 612,228 bp.

-Transcription of eccDNAs is not very prominent, yet a few anecdotal examples were found where split mapped reads across the breakpoint junctions were found in RNA-seq data. In figure 6 and example is shown for the HIP1 gene. Were the RNA reads found in exactly the same samples in which the circles were observed? And did the RNA split mapped reads for other circles also perfectly match the breakpoint observed at the DNA level? The authors could be more precise in the presentation of these data as this is important to judge their value.

Point-by-point response letter to reviewers

Attached is our point-by-point response to points raised by Reviewer #1, Reviewer #2, and Reviewer #3 with *Reviewers' comments in italic*.

Point-by-point response to Reviewer #1

Reviewer #1 (Remarks to the Author):

This is a much improved paper in which the authors have addressed all of the concerns which this reviewer had raised. The work is of general interest and will contribute to the long history of circular DNAs in higher eukaryotic cells. Exactly what mechanisms are involved in the generation of these circles however remains for future studies.

Response:

- We thank Reviewer #1 for the revision once again and appreciate the comments.

Two minor points: On line 323 the sentence ends without an ending.

Response:

- We are uncertain about this comment, as we could not find where the text should be changed.

On line 365 the authors state that circular DNA below 150 bp must be single stranded. This is quite wrong as there are ample examples of double stranded circles in the 70 bp size range (see papers of Charles Richardson in which mini circles are used as DNA templates. The authors just need to delete this comment.

Response:

- Thanks for your insightful knowledge.

Edits:

- The sentence "Their minute size may limit their transcription and eccDNA less than 150 bp must be expected to be entirely single-stranded DNA due to steric constraints¹" is now deleted.

Point-by-point response to Reviewer #2

Reviewer #2 (Remarks to the Author):

This reviewer continues to believe that this is a potentially important paper. However, the reviewers have still not fully addressed one of the key concerns, which is proving the existence of some of the larger gene containing eccDNAs using a non-PCR, non sequencing based methods, but rather providing some type of physical evidence. If the authors were to provide such evidence, or alternatively soften the claim if they can't do so, the paper would be stronger.

Response:

- We appreciate Reviewer #2 comments.
- In the discussion we reflect on this matter in **line 368-387**.
- We previously softened the claim that eccDNA larger than 25 kb exist in the text, by referring to the distance between breakpoints rather than to eccDNA.
- We have now revised the text additionally to be less conclusive.

Edits:

- Figure 3 has been changed so that titles are now "<25 kb" and ">25 kb". In lower part of figure 3 "eccDNA" has been changed to "EccDNA breakpoints" and the figure legend 3 was changed accordingly.
- On **line 368-371**, we soften our claim by rephrasing the sentence to: "*We detected more than a thousand different breakpoints more than 25 kb apart. Each of these putative circular DNA structures was detected on the basis of two independent pairs of structural read variants (Fig. 1c) that both supported the existence of a potential large eccDNA*".
- We have rewritten the section ("*Recurrent eccDNAs include full-length genes*") **line 226-250** softening our claim regarding larger genes that could be contained on eccDNAs and changed the title to: "*Recurrent breakpoints flank full-length genes*".

Point-by-point response to Reviewer #3

Reviewer #3 (Remarks to the Author):

This paper describes the identification of extrachromosomal circular DNA molecules in human muscle cells and leukocytes. The papers uses a previously established strategy for identification of eccDNA molecules, named Circle-seq, which essentially purifies and amplifies circular DNA. By paired-end sequencing the circular DNA molecule junction points are mapped.

A main concern appears the novelty of the findings. The authors and others have already published several papers on this topic, thus the identification of eccDNAs perse is not new.

The authors do extend previous work by describing somewhat larger eccDNAs up to 25 kb, showing transcription from eccDNAs and providing evidence for hotspots.

Response:

We thank Reviewer #3 for the revision.

Criticisms:

-title: the title suggests that the eccDNAs are part of the human genome, which is not the case, they are derived from chromosomal sequences but not part of it. I would suggest that the authors considering rephrasing this to e.g. Circular DNA elements of chromosomal origin are common in normal human cells. Or something along these lines

Response:

- Thanks for the comment.

Edits:

- We have changed the title to "*Circular DNA elements of chromosomal origin are common in healthy human somatic tissue*".

-results first section: please consider building this up in a better way. Now it is unclear how many unique eccDNA species were found, how many are recurring in independent samples/individuals, how much filtering was done to get to a robust collection of eccDNA sequences, how many were high and low conf, etc. Please try to be as transparent as possible.

Response:

- We agree that a better structure of the first result section will provide a better overview of our findings.
- Filtering excluded detection of coordinates mapped by a single structural read variant. Conditions for filtering are given in **line 110-120** and the bioinformatics pipeline is described in “*Mapping pipeline for Circle-Seq*” in Methods, **line 781-833**.
- All detected eccDNAs were hierarchically ranked (hconf, conf and lowq). Lists of all detected eccDNAs are provided Table S2 and Table S3, if researcher would like to do additional data filtering.
- Recurrence of eccDNAs between three samples from the same individual is clearly written in the paragraph “*EccDNA variation between cells in the same individual*”, **line 297-311**. Due to limited space, much more information can be found in extended data Figure 7+8+9 and Supplementary Table 10, if the reader would like to know more.
- Recurrence of eccDNAs between individuals is treated in the paragraph “*Recurrent breakpoints flank full-length genes*”, **line 226-250** and Figure 3. Description of recurrent eccDNAs is so extensive that we cannot merge these results into the first paragraph as suggested by the reviewer.

Edits:

- We have inserted the following sentence in **line 120-125**: “*In total, we detected 43,960 hconf eccDNAs, 81,066 conf eccDNAs and 13,655 lowq eccDNAs from muscle samples (average: 2,748 hconf, 5,067 conf and 853 lowq per sample of 10^6 nuclei. From leukocytes were detected 6,253 hconf eccDNAs, 3,191 conf eccDNAs and 784 lowq eccDNAs (average: 391 hconf, 199 conf, 49 lowq per sample of 10^4 nuclei) (Supplementary **Table 2**)*”.

-The classification of eccDNAs in hconf and conf and lowq categories appears somewhat subjective. At the very least the authors should perform experimental validation using e.g. PCR and Sanger sequencing to verify the presence of the eccDNA molecules. How this should be done is unclear though, as many sequences appear unique and can only be verified in amplified DNA. Perhaps the authors could consider a long-read sequencing platform to capture entire circles in reads (i.e. Nanopore sequencing).

Response:

- We validated 17 out of 20 detected eccDNAs (13 hconf and 7 conf) by outward PCR. Please see **Figure 1, Figure 4**, extended data **Figure 4, Supplementary Table 5** and M&M **line 923-940**.
- We thank the reviewer for suggesting Nanopore sequencing as an alternative method to identify and map eccDNA. However, we find it beyond the scope of this manuscript to develop another method for sequencing and mapping pipeline of eccDNA, when three methods have already been applied to verify eccDNA (Circle-Seq, recording mRNA from eccDNA break points, outward PCR and Sanger sequencing of PCR products across eccDNA break points).

Edits:

- The text in **line 207-213** was changed to: “We verified 85% of all tested eccDNAs (17 out of 20) by Sanger sequencing of outward PCR products (**Fig. 4a, Fig. 1d-f, Extended Data Fig. 5, Supplementary Table 5**). Tested unique eccDNAs came both from intergenic and genic loci”.

-An assessment of data quality was done by testing the NA12878 platinum genome dataset. The authors did not find rigorous support for eccDNAs in these data. Yet, I would suggest that such a dataset should also contain eccDNAs, that were extracted along with the chromosomal DNA when these data were generated. How do the authors explain the complete absence of eccDNAs in the NA12878 data?

Response:

- We agree that the whole-genome data could contain signals from eccDNAs. However, the signals from eccDNAs in normal whole-genome data are not enriched and could challenge our bioinformatics eccDNA detection due to high linear DNA content. Moreover, only a fraction of nuclei are expected to carry eccDNA. This result was actually shown earlier this year by Turner et al., 2017 *Science*.
- Furthermore, in the NA12878 dataset, we actually detected 7158 putative eccDNAs. However, these were supported solely by discordant paired-end reads with a majority 6661 (92.5%) deriving from repeat-masked regions (**Supplementary Table 2**). Extreme deep-sequencing could perhaps capture more junction reads to further support some of these putative detected eccDNAs.

Edits:

- We cannot exclude that some of the false positives described in **line 129-139** were actually signals from eccDNAs in NA12878 Platinum genome. We added the sentence “*In addition, it cannot be excluded that some of the signals interpreted as false positives in the NA12878 genome dataset is actually deriving from eccDNA in the whole genome data. Hence the rate of false positives might be lower*”.

Further, the authors later on state the presence of 7158 putative eccDNAs in NA12878. This is confusing and should be rephrased or presented in a more insightful way. E.g. consider showing the same statistics as for the muscle and leukocyte samples, i.e. visualize hconf, lowconf eccDNAs, perhaps normalized for the total amount of reads generated. Another point relates to the value of using NA12878 as a measure of specificity. Ideally, the authors should using Phi29 amplified genomic DNA for this. It is well known that different types of rearrangements (inversions/tandem duplications) are induced by Phi29 amplification, which could result in spurious split mappings.

Response:

- We agree that the false positive rate given by paired end reads mapping discordantly in the NA12878 genome can appear confusing.
- We are not convinced that Phi29 amplified genomic DNA provides an accurate false positive control for this study. Circular DNA was the primary substrate for Phi29 in the current study and not linear DNA. Hence, mutations, introduced by the proofreading Phi29 polymerase when applying linear DNA as substrate, might give an overestimate of false positives.

Edits:

- We have deleted the sentence: “*In the NA12878 dataset, we detected another 7158 putative eccDNAs but these were supported solely by discordant paired-end reads with a majority 6661 deriving from repeat-masked regions (sequences for interspersed repeats and low complexity DNA sequences, such as long interspersed nuclear elements; LINES, Supplementary Table 2).*”

-Another – in my view – important analysis lacking from the paper is a comparison of eccDNA junction points with known SVs breakpoint junctions in the human genome, such as those found in 1KG data. Is there any overlap between such datasets?

Response:

- As requested by Reviewer 3, we have now tested this for all detected eccDNA coordinates more than 25 kb apart, using a reciprocal overlap of 99% against available coordinates in the database of genomic variants².

Edits:

- To the text we added a new text section “Co-occurrence between eccDNAs and structural variants” in **line 251-262** “We further tested the co-occurrence between coordinates of common structural variants in the human genome² and eccDNA breakpoints > 25 kb apart. We found 22 hits with a reciprocal overlap of 99%. The hits included, among others, common deletions of genes from PRAMEF14/15/17/19/20 (220 kb, 1p36.21), DNAH14 (115 kb, 1q42.12), PSG4/5/9/10 (191 kb, 19q13.31) as well as repeats from satellites (1p11.2 and 5q11.1) and regions with LINE/SINE/LTR elements (131 kb, 1q12-1q21.1; 35 kb, 15q11.2). We also intersected a common inversion of genes encoding apolipoprotein L with the 54 kb [APO1_APO4^{circle}]. Missense variants of the APO1 gene is reported to be associated with a 15% increased risk of kidney disease³. Finally, a common deletion of the immunoglobulin heavy chain variable region (281 kb, 22q12.3) overlapped a detected eccDNA in sample T6: [abParts^{circle chr2: 89,161,023-89,441,956}].”
- We added a method section to material and methods, **line 898-900**: “Intersection with common genomic variants: Breakpoint coordinates of eccDNAs bigger than 25 kb were intersected against the database of genomic variants² with BedTools (v2.26.0-148-gd1953b6), using a reciprocal overlap of 99%.”

-Figure 1e: GD appears unexplained. Further, the T5 band was apparently not sequenced, while the main text states that both T5 and junctions were sequenced. Please clarify.

Response:

- Thanks for noticing.

Edits:

- To the Figure 1 legend was added: “and gel-image of T5 and T6 PCR products next to controls: GD, human genomic DNA; Ø, phi29 sample without detected [TTN^{circle}].”

-Line 152: how many lowq large eccDNAs were experimentally confirmed?

Response:

- In participant 4 (T4), we found a 35 kb long mRNA transcript matching a lowq detected 618 kb [TTN^{circle}] in T4, Supplementary **Table 9**. We did not run outward PCR tests for lowq eccDNAs but we did confirm 85% of all tested eccDNAs (17 of 20) by Sanger sequencing.

-Line 156: it appears almost impossible from figure 2A/B to find out if there is a difference between the active and inactive groups. How was lack of significance tested? Perhaps a power calculation would be

relevant in this context, to at least indicate the smallest difference which could still be picked up by comparing the data from these two groups.

Response:

- We agree this is difficult and we now present statistical data from a Wilcoxon rank sum test.

Edits:

- We report our statistical tests on **line 163-164** and **171**, showing no significance difference between the active and inactive groups (p-value = 0.95 and 0.43). Giving the wide confidence interval for muscle tissue samples (-7137 to 6074), we lowered our claim, and added a note, **line 164-166**, "*Although, given the wide confidence interval, significant differences between eccDNA counts from inactive and active men might have been missed with the current sample size (Fig. 2a and b).*".

-Line 161: EccDNA from...active and inactive men. This sentence mixes up two different things: the comparison in size distr between leukocytes and muscle, and (ii) differences between active and inactive men. Are these somehow related or are the authors just trying to squeeze two unrelated pieces of information in one sentence?

Response:

- We agree. The data was squeezed so we extended this section to make it clearer.

Edits:

- In **line 167-174**, the text was altered: "*EccDNA from the 16 leukocyte samples, each from approximately 10^4 nuclei (Extended Data Fig. 2f), showed comparable eccDNA frequencies and size distributions (Extended Data Fig. 4a-b, Supplementary Table 4) between physically active (median 356, range 284-919) and inactive men (median 528, range 218-2347), finding no significant difference based on a Wilcoxon rank sum test (p-value = 0.43, 95% confidence interval, -783 to 136). When comparing eccDNA frequencies from both types of examined soma, we detected 3.5 - 5.6 different eccDNAs/100 nuclei of leukocytes compared to 0.85 - 0.93 different eccDNAs/100 nuclei in muscle tissue.*".

-Plasmids were used as spike-in controls to measure the sensitivity of Circle-seq: The text describing these results appears rather imprecise. Was the sensitivity to be able to pick up circular DNA molecules 1/10,000 irrespective of circle size? Why is this semi-quantitative? The reported ranges appear very broad, how precise and reliable are these estimates?

Response:

- We do not understand the reviewer's comment. The paragraph **line 178 - 184** involves a clear statement that quantification is made for eccDNA in the size of 4 kb and elements in the size of SINEs.
- Our calculations and reasoning are explained in Methods **line 868-878**.
- Our estimates are found to be in agreement with previous findings that rely on purification of eccDNA with CsCl density gradient centrifugation, which is the closest we can get to evaluate how precise our estimates are.
- We agree that the word "semi-quantitative" is not justified.

Edits:

- We removed the word "semi-quantitative" and altered the text slightly in **line 178 - 184**.

-Line 180: the authors state that 0.8% of sequences were mapping to telomeres and conclude that TELcircles are therefore common in normal human cells. But how common is common in this case? 0.8% seems rather uncommon to me. And what is the evidence that these circles containing telomeric sequences are comparable to those described previously (refs 38-40)?

Response:

- We agree that the word “common” is a qualitative term.

Edit:

- We have removed the word “common” and altered the text to: “We found that 0.8% of reads mapped to telomeres, suggesting that [TEL^{circles}] also are present in healthy tissue”.

-Line 181: were the fractions of reads from circles that map to LINEs/SINEs/etc higher or lower than would be expected on the frequency of these elements in the human genome? I.e. was there over- or underrepresentation?

Response:

- This is an important question but not the focus of this paper.
- We will soon publish a second paper that focus on this question and where we have proper space to discuss and show this result.

-Fig4: The authors here attempted to verify several (large) circles with genes. It appears that not all functions were fully recovered in the Sanger sequencing the PCR products. Could the authors clarify the exact confirmation rate and, if the breakpoint junction was verified at nucleotide precision? The title of this section appears somewhat misplaced (lines 188-202) as the verification also involves many smaller circles.

Response:

- Thanks for pointing this out. We agree.

Edits:

- We changed the title and made two text sections instead of one with revised text, **line 207-224**.

-Line 262: 612,228 kb should probably be 612,228 bp.

Edits:

- Changed.

-Transcription of eccDNAs is not very prominent, yet a few anecdotal examples were found where split mapped reads across the breakpoint junctions were found in RNA-seq data. In figure 6 and example is shown for the HIP1 gene.

Response:

- We fully agree but please note our discussion of this “**line 350-354**”, where we state this is likely an underestimate.

-Were the RNA reads found in exactly the same samples in which the circles were observed? And did the RNA split mapped reads for other circles also perfectly match the breakpoint observed at the DNA level? The authors could be more precise in the presentation of these data as this is important to judge their

value.

Response:

- Yes, the junction overlap at the *HIP1* gene was a perfect match between RNA and DNA reads from the same participant (T8).
- In **Supplementary Table 9**, we provide the full list of structural read RNA variations that overlaps detected eccDNA regions inclusive size and number of read support. In **line 961-968** we describe the definition of our ranking based on mapping quality.

Edits:

- We updated figure 6 to include structural read variants, supporting the detected eccDNA on the DNA level.
- We added better description to the legend of **Figure 6**.
- We added information about participant T8 in the text, **line 287-290**.

References

1. Shore, D., Langowski, J. & Baldwin, R. L. DNA flexibility studied by covalent closure of short fragments into circles. *Proceedings of the National Academy of Sciences* **78**, 4833–4837 (1981).
2. MacDonald, J. R., Ziman, R., Yuen, R. K. C., Feuk, L. & Scherer, S. W. The Database of Genomic Variants: a curated collection of structural variation in the human genome. *Nucleic Acids Research* **42**, D986–92 (2014).
3. Dummer, P. D. *et al.* APOL1 Kidney Disease Risk Variants: An Evolving Landscape. *Semin. Nephrol.* **35**, 222–236 (2015).

Point-by-point response letter to reviewers

Attached is our point-by-point response to points raised by Reviewer #1, Reviewer #2, and Reviewer #3 with *Reviewers' comments in italic*.

Point-by-point response to Reviewer #1

Reviewer #1 (Remarks to the Author):

This is a much improved paper in which the authors have addressed all of the concerns which this reviewer had raised. The work is of general interest and will contribute to the long history of circular DNAs in higher eukaryotic cells. Exactly what mechanisms are involved in the generation of these circles however remains for future studies.

Response:

- We thank Reviewer #1 for the revision once again and appreciate the comments.

Two minor points: On line 323 the sentence ends without an ending.

Response:

- We are uncertain about this comment, as we could not find where the text should be changed.

On line 365 the authors state that circular DNA below 150 bp must be single stranded. This is quite wrong as there are ample examples of double stranded circles in the 70 bp size range (see papers of Charles Richardson in which mini circles are used as DNA templates. The authors just need to delete this comment.

Response:

- Thanks for your insightful knowledge.

Edits:

- The sentence “*Their minute size may limit their transcription and eccDNA less than 150 bp must be expected to be entirely single-stranded DNA due to steric constraints¹*” is now deleted.

Point-by-point response to Reviewer #2

Reviewer #2 (Remarks to the Author):

This reviewer continues to believe that this is a potentially important paper. However, the reviewers have still not fully addressed one of the key concerns, which is proving the existence of some of the larger gene containing eccDNAs using a non-PCR, non sequencing based methods, but rather providing some type of physical evidence. If the authors were to provide such evidence, or alternatively soften the claim if they can't do so, the paper would be stronger.

Response:

We appreciate Reviewer #2 comments.

We agree that it would be advantageous to have a PCR and sequencing independent proof of eccDNA larger than 100 kb. However, we are only aware of one method, electron microscopy (EM), which can address eccDNA size independent of PCR. EM has been used previously to prove the existence of rare eccDNA from somatic tissue in sizes up to 80 kb in mice. Larger eccDNAs, such as double minutes in tumors, have only been detected by EM when these have been selected for. We do therefore not expect to easily detect large and rare eccDNA in our samples from healthy muscle and leukocytes. Alternative methods for detection of Mb eccDNA could be fluorescence microscopy of metaphase cells as done for detection of ring chromosomes. However, large eccDNAs are much more rare than ring chromosomes and it will be difficult to distinguish them from chromosomes (noise) with this method in tissue and blood. We predict 1 eccDNA per 100 nucleus in sizes > 25 kb and even fewer for larger eccDNAs in leukocytes and muscle cells, where only few cells will be in the dividing metaphase where DNA can be visualized. We are in the process of developing a new and better method for detection of purified Mb eccDNA based on fluorescence and high resolution microscopy. However, such a method will not be ready within the next 6 month and thus, this is not within the scope of the current manuscript.

We have therefore followed Reviewer #2 alternative suggestion regarding large eccDNA:

2nd review: If the authors were to provide such evidence, or alternatively soften the claim if they can't do so, the paper would be stronger

by moderating our claims about eccDNA > 25 kb throughout the manuscript.

Edits:

Figure 3 has been changed so that titles are now "<25 kb" and ">25 kb". In lower part of figure 3 "eccDNA" has been changed to "EccDNA breakpoints" and the figure legend 3 was changed accordingly.

- Line 26: The word "eccDNA" was changed to "structures".
- Line 159: The sentence: "suggesting existence of eccDNA as large as 1 megabase" has been changed to suggesting potential existence of eccDNA as large as 1 megabase".
- Line 227– 250: We have rewritten the section "Recurrent eccDNAs include full-length genes", so that the wording "The majority of >25 kb eccDNAs" has been moderated to:
"The majority of breakpoints >25 kb apart "
- Line 368-371: "We discovered more than thousand different eccDNAs with breakpoints more than 25 kb apart. Each of these eccDNA was detected on the basis of two independent pairs of structural read variants (**Fig. 1c**) that both supported the existence of a potential large circular structure."

Has been changed to:

"We detected more than a thousand different breakpoints more than 25 kb apart. Each of these putative circular DNA structures was detected on the basis of two independent pairs of structural read variants (**Fig. 1c**) that both supported the existence of a potential large eccDNA".

Point-by-point response to Reviewer #3

Reviewer #3 (Remarks to the Author):

This paper describes the identification of extrachromosomal circular DNA molecules in human muscle cells and leukocytes. The paper uses a previously established strategy for identification of eccDNA molecules, named Circle-seq, which essentially purifies and amplifies circular DNA. By paired-end sequencing the circular DNA molecule junction points are mapped.

A main concern appears the novelty of the findings. The authors and others have already published several papers on this topic, thus the identification of eccDNAs per se is not new.

The authors do extend previous work by describing somewhat larger eccDNAs up to 25 kb, showing transcription from eccDNAs and providing evidence for hotspots.

Response:

We thank Reviewer #3 for the revision.

Criticisms:

-title: the title suggests that the eccDNAs are part of the human genome, which is not the case, they are derived from chromosomal sequences but not part of it. I would suggest that the authors considering rephrasing this to e.g. Circular DNA elements of chromosomal origin are common in normal human cells. Or something along these lines

Response:

- Thanks for the comment.

Edits:

- We have changed the title to “Circular DNA elements of chromosomal origin are common in healthy human somatic tissue”.

-results first section: please consider building this up in a better way. Now it is unclear how many unique eccDNA species were found, how many are recurring in independent samples/individuals, how much filtering was done to get to a robust collection of eccDNA sequences, how many were high and low conf, etc. Please try to be as transparent as possible.

Response:

- We agree that a better structure of the first result section will provide a better overview of our findings.
- Filtering excluded detection of coordinates mapped by a single structural read variant. Conditions for filtering are given in **line 110-120** and the bioinformatics pipeline is described in “Mapping pipeline for Circle-Seq” in Methods, **line 781-833**.
- All detected eccDNAs were hierarchically ranked (hconf, conf and lowq). Lists of all detected eccDNAs are provided Table S2 and Table S3, if researcher would like to do additional data filtering.
- Recurrence of eccDNAs between three samples from the same individual is clearly written in the paragraph “EccDNA variation between cells in the same individual”, **line 297-311**. Due to limited space, much more information can be found in extended data Figure 7+8+9 and Supplementary Table 10, if the reader would like to know more.

- Recurrence of eccDNAs between individuals is treated in the paragraph "*Recurrent breakpoints flank full-length genes*", **line 226-250** and Figure 3. Description of recurrent eccDNAs is so extensive that we cannot merge these results into the first paragraph as suggested by the reviewer.

Edits:

- We have inserted the following sentence in **line 120-125**: "*In total, we detected 43,960 hconf eccDNAs, 81,066 conf eccDNAs and 13,655 lowq eccDNAs from muscle samples (average: 2,748 hconf, 5,067 conf and 853 lowq per sample of 10^6 nuclei. From leukocytes were detected 6,253 hconf eccDNAs, 3,191 conf eccDNAs and 784 lowq eccDNAs (average: 391 hconf, 199 conf, 49 lowq per sample of 10^4 nuclei) (Supplementary **Table 2**)*".

-The classification of eccDNAs in hconf and conf and lowq categories appears somewhat subjective. At the very least the authors should perform experimental validation using e.g. PCR and Sanger sequencing to verify the presence of the eccDNA molecules. How this should be done is unclear though, as many sequences appear unique and can only be verified in amplified DNA. Perhaps the authors could consider a long-read sequencing platform to capture entire circles in reads (i.e. Nanopore sequencing).

Response:

- We validated 17 out of 20 detected eccDNAs (13 hconf and 7 conf) by outward PCR. Please see **Figure 1, Figure 4**, extended data **Figure 4, Supplementary Table 5** and M&M **line 923-940**.
- We thank the reviewer for suggesting Nanopore sequencing as an alternative method to identify and map eccDNA. However, we find it beyond the scope of this manuscript to develop another method for sequencing and mapping pipeline of eccDNA, when three methods have already been applied to verify eccDNA (Circle-Seq, recording mRNA from eccDNA break points, outward PCR and Sanger sequencing of PCR products across eccDNA break points).

Edits:

- The text in **line 207-213** was changed to: "*We verified 85% of all tested eccDNAs (17 out of 20) by Sanger sequencing of outward PCR products (Fig. 4a, Fig. 1d-f, Extended Data Fig. 5, Supplementary Table 5). Tested unique eccDNAs came both from intergenic and genic loci*".

-An assessment of data quality was done by testing the NA12878 platinum genome dataset. The authors did not find rigorous support for eccDNAs in these data. Yet, I would suggest that such a dataset should also contain eccDNAs, that were extracted along with the chromosomal DNA when these data were generated. How do the authors explain the complete absence of eccDNAs in the NA12878 data?

Response:

- We agree that the whole-genome data could contain signals from eccDNAs. However, the signals from eccDNAs in normal whole-genome data are not enriched and could challenge our bioinformatics eccDNA detection due to high linear DNA content. Moreover, only a fraction of nuclei are expected to carry eccDNA. This result was actually shown earlier this year by Turner et al., 2017 *Science*.
- Furthermore, in the NA12878 dataset, we actually detected 7158 putative eccDNAs. However, these were supported solely by discordant paired-end reads with a majority 6661 (92.5%) deriving from repeat-masked regions (Supplementary **Table 2**). Extreme deep-sequencing could perhaps capture more junction reads to further support some of these putative detected eccDNAs.

Edits:

- We cannot exclude that some of the false positives described in **line 129-139** were actually signals from eccDNAs in NA12878 Platinum genome. We added the sentence “*In addition, it cannot be excluded that some of the signals interpreted as false positives in the NA12878 genome dataset is actually deriving from eccDNA in the whole genome data. Hence the rate of false positives might be lower*”.

Further, the authors later on state the presence of 7158 putative eccDNAs in NA12878. This is confusing and should be rephrased or presented in a more insightful way. E.g. consider showing the same statistics as for the muscle and leukocyte samples, i.e. visualize hconf, lowconf eccDNAs, perhaps normalized for the total amount of reads generated. Another point relates to the value of using NA12878 as a measure of specificity. Ideally, the authors should use Phi29 amplified genomic DNA for this. It is well known that different types of rearrangements (inversions/tandem duplications) are induced by Phi29 amplification, which could result in spurious split mappings.

Response:

- We agree that the false positive rate given by paired end reads mapping discordantly in the NA12878 genome can appear confusing.
- We are not convinced that Phi29 amplified genomic DNA provides an accurate false positive control for this study. Circular DNA was the primary substrate for Phi29 in the current study and not linear DNA. Hence, mutations, introduced by the proofreading Phi29 polymerase when applying linear DNA as substrate, might give an overestimate of false positives.

Edits:

- We have deleted the sentence: “*In the NA12878 dataset, we detected another 7158 putative eccDNAs but these were supported solely by discordant paired-end reads with a majority 6661 deriving from repeat-masked regions (sequences for interspersed repeats and low complexity DNA sequences, such as long interspersed nuclear elements; LINEs, Supplementary **Table 2**).*”

-Another – in my view – important analysis lacking from the paper is a comparison of eccDNA junction points with known SVs breakpoint junctions in the human genome, such as those found in 1KG data. Is there any overlap between such datasets?

Response:

- As requested by Reviewer 3, we have now tested this for all detected eccDNA coordinates more than 25 kb apart, using a reciprocal overlap of 99% against available coordinates in the database of genomic variants².

Edits:

- To the text we added a new text section “*Co-occurrence between eccDNAs and structural variants*” in **line 251-262** “*We further tested the co-occurrence between coordinates of common structural variants in the human genome² and eccDNA breakpoints > 25 kb apart. We found 22 hits with a reciprocal overlap of 99%. The hits included, among others, common deletions of genes from PRAMEF14/15/17/19/20 (220 kb, 1p36.21), DNAH14 (115 kb, 1q42.12), PSG4/5/9/10 (191 kb, 19q13.31) as well as repeats from satellites (1p11.2 and 5q11.1) and regions with LINE/SINE/LTR elements (131 kb, 1q12-1q21.1; 35 kb, 15q11.2). We also intersected a common inversion of genes encoding apolipoprotein L with the 54 kb [APO1_APO4^{circle}]. Missense variants of the APO1 gene is reported to be associated with a 15% increased risk of kidney disease³. Finally, a common deletion of the immunoglobulin heavy chain variable region (281 kb, 22q12.3) overlapped a detected eccDNA in sample T6:*

[*abParts*^{circle chr2: 89,161,023-89,441,956}].”

- We added a method section to material and methods, **line 898-900**: “*Intersection with common genomic variants: Breakpoint coordinates of eccDNAs bigger than 25 kb were intersected against the database of genomic variants² with BedTools (v2.26.0-148-gd1953b6), using a reciprocal overlap of 99%.*”.

-*Figure 1e: GD appears unexplained. Further, the T5 band was apparently not sequenced, while the main text states that both T5 and junctions were sequenced. Please clarify.*

Response:

- Thanks for noticing.

Edits:

- To the Figure 1 legend was added: “*and gel-image of T5 and T6 PCR products next to controls: GD, human genomic DNA; Ø, phi29 sample without detected [*TTN*^{circle}].*”

-*Line 152: how many lowq large eccDNAs were experimentally confirmed?*

Response:

- In participant 4 (T4), we found a 35 kb long mRNA transcript matching a lowq detected 618 kb [*TTN*^{circle}] in T4, Supplementary **Table 9**. We did not run outward PCR tests for lowq eccDNAs but we did confirm 85% of all tested eccDNAs (17 of 20) by Sanger sequencing.

-*Line 156: it appears almost impossible from figure 2A/B to find out if there is a difference between the active and inactive groups. How was lack of significance tested? Perhaps a power calculation would be relevant in this context, to at least indicate the smallest difference which could still be picked up by comparing the data from these two groups.*

Response:

- We agree this is difficult and we now present statistical data from a Wilcoxon rank sum test.

Edits:

- We report our statistical tests on **line 163-164** and **171**, showing no significance difference between the active and inactive groups (p-value = 0.95 and 0.43). Giving the wide confidence interval for muscle tissue samples (-7137 to 6074), we lowered our claim, and added a note, **line 164-166**, “*Although, given the wide confidence interval, significant differences between eccDNA counts from inactive and active men might have been missed with the current sample size (Fig. 2a and b).*”.

-*Line 161: EccDNA from...active and inactive men. This sentence mixes up two different things: the comparison in size distr between leukocytes and muscle, and (ii) differences between active and inactive men. Are these somehow related or are the authors just trying to squeeze two unrelated pieces of information in one sentence?*

Response:

- We agree. The data was squeezed so we extended this section to make it clearer.

Edits:

- In **line 167-174**, the text was altered: “*EccDNA from the 16 leukocyte samples, each from approximately 10^4 nuclei (Extended Data Fig. 2f), showed comparable eccDNA frequencies and size distributions (Extended Data Fig. 4a-b, Supplementary Table 4) between physically active (median 356, range 284-919) and inactive men (median 528, range 218-2347), finding no significant difference based on a Wilcoxon rank sum test (p -value = 0.43, 95% confidence interval, -783 to 136). When comparing eccDNA frequencies from both types of examined soma, we detected 3.5 - 5.6 different eccDNAs/100 nuclei of leukocytes compared to 0.85 - 0.93 different eccDNAs/100 nuclei in muscle tissue.*”.

-Plasmids were used as spike-in controls to measure the sensitivity of Circle-seq: The text describing these results appears rather imprecise. Was the sensitivity to be able to pick up circular DNA molecules 1/10,000 irrespective of circle size? Why is this semi-quantitative? The reported ranges appear very broad, how precise and reliable are these estimates?

Response:

- We do not understand the reviewer’s comment. The paragraph **line 178 - 184** involves a clear statement that quantification is made for eccDNA in the size of 4 kb and elements in the size of SINEs.
- Our calculations and reasoning are explained in Methods **line 868-878**.
- Our estimates are found to be in agreement with previous findings that rely on purification of eccDNA with CsCl density gradient centrifugation, which is the closest we can get to evaluate how precise our estimates are.
- We agree that the word “semi-quantitative” is not justified.

Edits:

- We removed the word “semi-quantitative” and altered the text slightly in **line 178 - 184**.

-Line 180: the authors state that 0.8% of sequences were mapping to telomeres and conclude that TELcircles are therefore common in normal human cells. But how common is common in this case? 0.8% seems rather uncommon to me. And what is the evidence that these circles containing telomeric sequences are comparable to those described previously (refs 38-40)?

Response:

- We agree that the word “common” is a qualitative term.

Edit:

- We have removed the word “common” and altered the text to: “*We found that 0.8% of reads mapped to telomeres, suggesting that [TEL^{circles}] also are present in healthy tissue*”.

-Line 181: were the fractions of reads from circles that map to LINEs/SINEs/etc higher or lower than would be expected on the frequency of these elements in the human genome? I.e. was there over- or underrepresentation?

Response:

- This is an important question but not the focus of this paper.
- We will soon publish a second paper that focus on this question and where we have proper space to discuss and show this result.

-Fig4: The authors here attempted to verify several (large) circles with genes. It appears that not all functions were fully recovered in the Sanger sequencing the PCR products. Could the authors clarify the exact confirmation rate and, if the breakpoint junction was verified at nucleotide precision? The title of this section appears somewhat misplaced (lines 188-202) as the verification also involves many smaller circles.

Response:

- Thanks for pointing this out. We agree.

Edits:

- We changed the title and made two text sections instead of one with revised text, **line 207-224**.

-Line 262: 612,228 kb should probably be 612,228 bp.

Edits:

- Changed.

-Transcription of eccDNAs is not very prominent, yet a few anecdotal examples were found where split mapped reads across the breakpoint junctions were found in RNA-seq data. In figure 6 and example is shown for the HIP1 gene.

Response:

- We fully agree but please note our discussion of this “**line 350-354**”, where we state this is likely an underestimate.

-Were the RNA reads found in exactly the same samples in which the circles were observed? And did the RNA split mapped reads for other circles also perfectly match the breakpoint observed at the DNA level? The authors could be more precise in the presentation of these data as this is important to judge their value.

Response:

- Yes, the junction overlap at the *HIP1* gene was a perfect match between RNA and DNA reads from the same participant (T8).
- In **Supplementary Table 9**, we provide the full list of structural read RNA variations that overlaps detected eccDNA regions inclusive size and number of read support. In **line 961-968** we describe the definition of our ranking based on mapping quality.

Edits:

- We updated figure 6 to include structural read variants, supporting the detected eccDNA on the DNA level.
- We added better description to the legend of **Figure 6**.
- We added information about participant T8 in the text, **line 287-290**.

References

1. Shore, D., Langowski, J. & Baldwin, R. L. DNA flexibility studied by covalent closure of short fragments into circles. *Proceedings of the National Academy of Sciences* **78**, 4833–4837 (1981).
2. MacDonald, J. R., Ziman, R., Yuen, R. K. C., Feuk, L. & Scherer, S. W. The Database of Genomic

- Variants: a curated collection of structural variation in the human genome. *Nucleic Acids Research* **42**, D986–92 (2014).
3. Dummer, P. D. *et al.* APOL1 Kidney Disease Risk Variants: An Evolving Landscape. *Semin. Nephrol.* **35**, 222–236 (2015).

Reviewer #2 (Remarks to the Author):

The authors have done a good job of addressing the critiques. The paper is thought-provoking and will be of broad interest to the field.

Reviewer #3 (Remarks to the Author):

The authors have addressed most of my concerns. I have a few minor remaining suggestions.

line 212 - "...receptor 2 that are associated..." should be "...receptor 2 that is associated..."

line 875 - "... eccDNA is bias..." should be "...eccDNA is biased...". Furthermore, this whole sentence and the preceding sentence ("This estimate....phi29 polymerase" and "Considering the lowest...in sizes of 4 kb") is weirdly phrased. Perhaps this paragraph explains how spike-in plasmids were used to estimate eccDNA amounts per nucleus, but then at the very least the phrasing should be crystal clear and grammatically correct

Point-by-point response letter to reviewers

Attached is our point-by-point response to points raised by Reviewers with their comments in *italic*.

Reviewer #2 (Remarks to the Author):

The authors have done a good job of addressing the critiques. The paper is thought-provoking and will be of broad interest to the field.

Response:

- We thank Reviewer #2 for the comments.

Reviewer #3 (Remarks to the Author): The authors have addressed most of my concerns. I have a few minor remaining suggestions.

Response:

- Thanks.

line 212 - "...receptor 2 that are associated..." should be "...receptor 2 that is associated..."

Edit response:

- Corrected.

line 875 - "... eccDNA is bias..." should be "...eccDNA is biased...". Furthermore, this whole sentence and the preceding sentence ("This estimate...phi29 polymerase" and "Considering the lowest...in sizes of 4 kb") is weirdly phrased. Perhaps this paragraph explains how spike-in plasmids were used to estimate eccDNA amounts per nucleus, but then at the very least the phrasing should be crystal clear and grammatically correct.

Edit response:

- Corrected and changed to:
Quantification of eccDNA based on internal controls: The number of eccDNA per nucleus was calculated based on fractions of reads mapped to 4-kb spike-in plasmid controls (pUG72 and pBR322, **Fig. 2c**). Added to each muscle sample was 50,000 pUG72 and 20,000 pBR322 plasmids. Using the lowest and highest percent read values of pUG72 (0.0015% and 0.2596%) and pBR322 (0.0002% and 0.0905%), we estimated around 1 to 250 eccDNAs of 4 kb per nucleus. Using percent reads for SINE elements (1.676% and 8.432%) relative to percent reads for pUG72 or pBR322 per nucleus, we estimated [*SINE*^{circles}] of 16 to 1700 per nucleus with sizes of approximately 400 bp. These calculations are merely estimates because rolling-circle amplification of eccDNAs using phi29 polymerase is biased towards abundant and small eccDNAs⁶².